# Spectroscopic Properties of Erbium-Doped Oxyfluoride Phospho-Tellurite Glass and Transparent Glass-Ceramic Containing BaF_2_ Nanocrystals

**DOI:** 10.3390/ma12203429

**Published:** 2019-10-20

**Authors:** Magdalena Lesniak, Jacek Zmojda, Marcin Kochanowicz, Piotr Miluski, Agata Baranowska, Gabriela Mach, Marta Kuwik, Joanna Pisarska, Wojciech A. Pisarski, Dominik Dorosz

**Affiliations:** 1Faculty of Materials Science and Ceramics, AGH University of Science and Technology, Av. Mickiewicza 30, 30059 Krakow, Poland; machgabriela1@gmail.com (G.M.); ddorosz@agh.edu.pl (D.D.); 2Faculty of Electrical Engineering, Bialystok University of Technology, Wiejska Street 45D, 15351 Bialystok, Poland; j.zmojda@pb.edu.pl (J.Z.); m.kochanowicz@pb.edu.pl (M.K.); p.miluski@pb.edu.pl (P.M.); 3Faculty of Mechanical Engineering, Bialystok University of Technology, Wiejska Street 45C, 15351 Bialystok, Poland; 4Institute of Chemistry, University of Silesia, Szkolna 9, 40007 Katowice, Poland; marta.soltys@us.edu.pl (M.K.); joanna.pisarska@us.edu.pl (J.P.); wojciech.pisarski@us.edu.pl (W.A.P.)

**Keywords:** oxyfluoride glass, glass-ceramic, BaF_2_ nanocrystals, Er^3+^ co-doping, structure, spectroscopic properties, 2.7 μm emission, upconversion

## Abstract

The ErF_3_-doped oxyfluoride phospho-tellurite glasses in the (40-x) TeO_2_-10P_2_O_5_-45 (BaF_2_-ZnF_2_) -5Na_2_O-xErF_3_ system (where x = 0.25, 0.50, 0.75, 1.00, and 1.25 mol%) have been prepared by the conventional melt-quenching method. The effect of erbium trifluoride addition on thermal, structure, and spectroscopic properties of oxyfluoride phospho-tellurite precursor glass was studied by differential scanning calorimetry (DSC), Fourier-transform infrared (FTIR), and Raman spectroscopy as well as emission measurements, respectively. The DSC curves were used to investigate characteristic temperatures and thermal stability of the precursor glass doped with varying content of ErF_3_. FTIR and Raman spectra were introduced to characterize the evolution of structure and phonon energy of the glasses. It was found that the addition of ErF_3_ up to 1.25 mol% into the chemical composition of phospho-tellurite precursor glass enhanced 2.7 µm emission and upconversion. By controlled heat-treatment process of the host glass doped with the highest content of erbium trifluoride (1.25 mol%), transparent erbium-doped phospho-tellurite glass-ceramic (GC) was obtained. X-ray diffraction analysis confirmed the presence of BaF_2_ nanocrystals with the average 16 nm diameter in a glass matrix. Moreover, MIR, NIR, and UC emissions of the glass-ceramic were discussed in detail and compared to the spectroscopic properties of the glass doped with 1.25 mol% of ErF_3_ (the base glass).

## 1. Introduction

Since Wang reported for the first time in the literature the method of obtaining transparent oxyfluoride glass-ceramic doped with Er^3+^ and Yb^3+^ ions [1], the dynamic development of research on transparent rare-earth-doped (RED) oxyfluoride glass-ceramics (GC) with fluoride nanocrystals has been noted [2,3,4,5]. As a result of controlled heat treatment of the precursor glass in the SiO_2_-Al_2_O_3_-PbF_2_-CdF_2_-YbF_3_-ErF_3_ system, Wang and Ohwaki obtained Pb_x_Cd_1-x_F_2_ nanocrystals. The authors proved that the presence of a fluoride phase in the glass matrix increased the efficiency of upconversion with the participation of Er^3+^ and Yb^3+^ co-dopants [1].

Many advantages of fluoride crystals, such as the transparency from ultraviolet to infrared region, a wide range of the energy band gap (greater than 10 eV) and the low phonon energy (300–500 cm^−1^) make them active centers for the optical rare-earth ions in the glass-ceramic materials [2,3,4,5,6]. 

The glass-ceramics can be prepared by controlled crystallization of the glasses using different processing methods [7], but still, the common method to obtain GC is crystallization by a controlled heat treatment process. The RED glass-ceramics with nanocrystals in the glass matrix can be obtained as a result of controlled heat treatment above the glass transition temperature of oxide [8,9,10] and oxyfluoride glasses [11,12]. In this method, the rare-earth ions are incorporated into the nanocrystals by the diffusion-controlled process, that is dependent on the temperature. According to the literature [11,12], to obtain the optimum dopant concentration in the nanocrystals and thus the highest luminescence efficiency, the precise control of the rare-earth concentration and crystallization process is required. 

Transparent glass-ceramic is a very attractive material for mid-infrared (MIR) application because an active crystal phase in the glass matrix can be obtained [13,14,15]. Among the rare-earth ions used as optically active dopants in the glasses and GC, erbium ion can be recommended as a candidate for 3 µm emission due to the transition of ^4^I_11/2_→^4^I_13/2_ [13,14]. The tellurite glasses show the broadband transmission window (0.35–6 um), the high linear and nonlinear refractive indices, the low phonon energy (~750 cm^−1^) and the high rare-earth solubility. By the addition of suitable metal fluorides with low phonon energy, oxyfluoride tellurite glasses are expected to reduce the possibility of non-radiative transition, which can provide efficient emission and ensure a broad bandwidth [16,17].

Since the discovery of transparent GC in the K_2_O-Nb_2_O_5_-TeO_2_ system by Shioya et al. [18], tellurite GC has a special scientific interest. These materials combine the high mechanical stability and the high solubility of rare-earth ions of tellurite glass and have a low-phonon-energy environment of the precipitated nanocrystals. The low melt (~900 °C) and crystallization temperatures of the tellurite glasses allowed to easily control the crystal size of glass-ceramic. These advantages make tellurite glass-ceramic as a suitable glass matrix for MIR materials [19]. 

Not only oxyfluoride tellurite glass-ceramic but also oxyfluoride phosphate glass-ceramic is of particular interest as the hosts for luminescent rare-earth ions. This has been confirmed among others by publications of Petit et al. [20,21,22,23]. Adding P_2_O_5_ glass former to the tellurite glass improves its thermal stability and tensile strength. This is due to the network-former mixing effect in the tellurite glasses [24]. The addition of fluorine compounds into the phospho-tellurite glasses can provide important advantages compared to the pure fluoride and pure oxide glasses [24]. Oxyfluoride phospho-tellurite glasses combine the optical properties of fluoride and tellurite glasses with better mechanical and thermal stability of phosphate glasses [25]. 

Unfortunately, only a few studies have been conducted on oxyfluoride tellurite glass-ceramics [26,27,28]. Rajesh and Camargo [26] proposed oxyfluoride tellurite glasses and glass-ceramics containing NaYF_4_ nanocrystals doped with variable concentration of Nd^3+^ in the TeO_2_-ZnO-YF_3_-NaF-xNd_2_O_3_ (x = 0.2, 0.5, 1.0, 1.5, and 2.0 mol%) system for 1.06 µm emission. Transparent oxyfluoride glass-ceramics with SrF_2_ nanocrystals have been prepared in the system of TeO_2_-BaO-Bi_2_O_3_-SrF_2_-2RE_2_O_3_ (RE = Eu, Dy) [27]. Transparent oxyfluoride tellurite glasses with the composition of TeO_2_-SiO_2_-AlF_3_-CaO-NaF-yHoF_3_ (*y* = 0.5, 1, 1.5, 2.0, 3.0) were synthesized by Hou et al. [28]. The luminescence intensity of the glass and transparent glass-ceramics with β-NaCaAlF_6_ nanocrystals at 1.53 μm dropped monotonously with the increase of Ho^3+^ ions concentration under 980 nm LD excitation [28]. 

Definitely more reports in the literature on oxyfluoride glass-ceramic refers to the silicate glasses. Glass-ceramics prepared by the heat-treated glass from SiO_2_-Na_2_CO_3_-Al_2_O_3_-NaF-LuF_3_-Yb_2_O_3_-Er_2_O_3_ system was characterized by strong green upconversion fluorescence due to Er^3+^ doping in Na_5_Lu_9_F_32_ crystallites [29]. The novel nanostructured NdF_3_ glass-ceramic was obtained by Zhang et al. upon heating of the precursor glass with the composition of 50SiO_2_-22Al_2_O_3_-10NdF_3_-18NaF (in mol%) as the bandpass color filter for wide-color-gamut white LED [30]. Another example of glass-ceramics obtained from the precursor silicate glass can be Tb^3+^-doped transparent oxyfluoride glass-ceramics containing LiYF_4_ [31]. According to the authors, this material may be a novel scintillator applied for X-ray detection for the slow event [31]. 

The reports on the transparent oxyfluoride nano-glass-ceramics obtained by the annealing process of the silicate glasses can be also found in numerous publications of Durán et al. [11,12,32,33,34]. The results obtained by Duran et al. confirmed the incorporation of rare-earth in the nanocrystals (fluoride compounds) and explained their higher emission efficiency due to the lower phonon energy of the fluoride crystal lattice, which reduces the multi-phonon relaxation rates. 

The present work deals with the role of ErF_3_ on the thermal, structure, and near-, mid-infrared as well as the upconversion emission of host oxyfluoride phospho-tellurite glass. To our best knowledge, no work is reported about oxyfluoride tellurite glasses containing 45 mol% of fluorine compounds in the host glass. In our study, we proposed the host oxyfluoride phospho-tellurite glass with ca. 46 mol% of fluoride compounds. After obtaining the chemically and thermally stable glass, we prepared the glass-ceramic with BaF_2_ nanocrystals by controlling the heat treatment. No reports of the transparent Er^3+^-doped BaF_2_ glass-ceramic in the oxyfluoride phospho-tellurite system exist in the literature. Comparing to the precursor glass, we achieved an enhanced 2.7 µm emission and upconversion from erbium-doped glass-ceramic upon excitation with 976 nm laser diode due to the incorporation of erbium ions into the BaF_2_ nanocrystals.

## 2. Experimental Procedure

Oxyfluoride phospho-tellurite glasses with the molar composition (40-x)TeO_2_-10P_2_O_5_-45(BaF_2_-ZnF_2_)-5Na_2_O-xErF_3_ system (where x = 0.25, 0.50, 0.75, 1.00, and 1.25 mol%), denoted as TP0.25ErF_3_, TP0.50ErF_3_, TP0.75ErF_3_, TP1.00ErF_3_, TP1.25ErF_3_, respectively, were prepared by conventional melt-quenching method. High purity (99.99%) raw materials (TeO_2_, P_2_O_5_, BaF_2_, ZnF_2_, Na_2_CO_3_, and ErF_3_) were used and each batch of 10 g was well-mixed in an agate mortar, and then melted in a covered platinum crucible at 950 °C for 90 min in ambient air. The melt was cast onto a preheated stainless steel, next annealed at 310 °C for 10 h, and then cooled down to room temperature. All the obtained samples were transparent. The glass samples were cut to the 5 mm × 5 mm size and polished to optical quality before measurements with a 2 mm thickness. Crystallized material was obtained for the addition of 1.5 mol% ErF_3_. Therefore, the glass with 1.25 mol% ErF_3_ content has been chosen to obtain a transparent glass-ceramic sample. Based on the recorded DSC curve of the TP1.25ErF_3_ glass, it was found that the onset crystallization temperature was 390 °C and the significant exothermic crystal peak corresponded to 456 °C. According to this data, the heat treatment temperature of TP1.25ErF_3_ glass sample was determined from 395 to 460 °C. In order to obtain the BaF_2_ fluoride crystalline phase in a nanometric scale, the temperature of the controlled heat treatment process of TP1.25ErF_3_ glass was set at 400 °C for 3 h.

The sample with 1.25 mol% of erbium trifluoride was heat-treated for 3 h at 400 °C to obtain the glass-ceramic.

X-ray diffraction studies were carried out on the X’Pert Pro X-ray diffractometer supplied by PANalytical (Almelo, Netherlands) with Cu K_α1_ radiation (λ = 1.54056 Å) in the 2θ range of 10–90°. The step size, time per step, and scan speed were as follows: 0.017°, 184.79 s, and 0.011°/s. The X-ray tube was operated at 40 kV and 40 mA and a scintillation detector (Almelo, Netherlands) were used to measure the intensity of the scattered X-rays. Qualitative identification of the phase composition in the glass-ceramic sample was performed with reference to the ICDD PDF-2 database. From the peak width of the X-ray pattern of the glass-ceramic sample, the crystalline size of crystals can be calculated on the basis of Scherrer‘s equation. 

The thermal characteristic temperatures such as the glass transition (T_g_), the crystallization (T_c_), and melting (T_m_) temperatures were measured by using the Jupiter DTA STA 449 F3 thermal analyzer (NETZSCH Thermal Analysis, Selb, Germany), operating in the heat flux DSC mode at a heating rate of 10 °C/min under synthetic air atmosphere. Measurements were carried out with an uncertainty of ±1 °C.

The FTIR spectra of the glasses were obtained with the Fourier spectrometer (Bruker Optics-Vertex70V, Rheinstetten, Germany). The measurements were done using the KBr pellet technique. Absorption spectra were recorded at 128 scans and the resolution of 4 cm^−1^. 

Raman spectra of all samples were obtained using a LabRAM HR spectrometer (HORIBA Jobin Yvon, Palaiseau, France) using the excitation wavelength of 532 nm. The diffraction grating was 1800 lines/mm. The spectra were recorded in several points with the standard spot of about 1 μm. 

Raman and FTIR spectra have been normalized and then deconvoluted using Fityk software (0.9.8 software, open-source (GPL2+)). The coefficient of determination (R square) of all the deconvoluted FTIR and Raman spectra was 0.99. The standard deviation of the position and full width at half maximum (FWHM) of each of the component bands was ± 4 cm^−1^. 

The mid-infrared spectra were obtained with the Acton Spectra Pro 2300i monochromator (Princeton Instruments, Trenton, NJ, USA) in the spectral range of 2550–2850 nm using high power laser diode (λ_exc_ = 976 nm) as a pump source and PbS detector supported by lock-in-amplifier SR510 (Stanford Research Systems, Sunnyvale, CA, USA). The NIR luminescence was performed using the Acton Spectra Pro 2300i monochromator with InGaAs detector in the range of 1.4–1.7 μm. Both measurements were carried in the transmission mode, where the laser beam was focused on the surface of the glass sample and the luminescence signal was collected on the entrance of monochromator. The upconversion luminescence spectra of the glasses and glass-ceramic were measured in a range of 500–700 nm using the Stelarnet GreenWave monochromator (Stellarnet Inc., Tampa, FL, USA) and the laser diode (λ_exc_ = 976 nm). In this part of the experiment the luminescence signal was collected by transmitting optical fiber with NA = 0.5 and 400 µm diameter. The fiber end was located perpendicular to the excitation laser beam. All the measurements were carried out at room temperature. In order to eliminate the strong laser radiation, spectral filters FEL 1250 (Thorlabs, NJ, USA) for NIR and MIR spectra measurements and “heat glass” in the case of UC luminescence were used. 

## 3. Results

### 3.1. Studies of Erbium-Doped Oxyfluoride Phospho-Tellurite Glass 

#### 3.1.1. X-ray Diffraction Analysis

Figure 1 presents X-ray diffraction patterns of the erbium-doped phospho-tellurite glass with the varying content of ErF_3_ (from 0.25 to 1.25 mol%). The observed diffraction patterns showed the amorphous character of all the samples. The diffractograms confirmed the absence of any crystalline phase, only broad and intense broad humps in the 20–35° range of the two theta angles are observed. X-ray diffraction analysis of the studied glasses shows that all oxyfluoride phospho-tellurite glasses with different contents of ErF_3_ exhibits similar patterns and the mentioned hump does not become broader with the increasing erbium fluoride content. It suggests the absence of the evolution to a lower degree of the order of the local structure [35].

#### 3.1.2. Thermal Properties of Oxyfluoride Erbium-Doped Phospho-Tellurite Glass

Thermal stability factor of the glass ΔT, defined as a resistance to crystallization during heating, is an important parameter to consider in the manufacture and technological applications of the glass. Thermal stability factor is measured as the difference between the crystallization T_c_ and the glass transition T_g_ temperatures. If the value of ΔT is higher than 100 °C, it is assumed that the glass can be considered as thermally stable [36].

Tellurite glasses combine both useful technological properties such as low melting temperatures and good thermal stability [36,37]. The phospho-tellurite glasses formed chemically stable glasses over wide compositional ranges [38,39,40,41,42]. According to the results of Rinke et al., the addition of TeO_2_ to sodium phosphate glass caused the increase in the glass transition temperature from 284 to 327 °C [43]. The increase in the glass transition temperature with the addition of P_2_O_5_ into the chemical composition of tellurite glass was also observed by Nandi and Jose [44]. Phosphorus pentoxide (P_2_O_5_) also plays a role as the stabilizing component, influencing the structure and the crystallization behavior in fluoride systems [45,46,47]. The influence of P_2_O_5_ on the thermal properties of fluoroindate glasses activated by Pr^3+^ ions was reported by Pisarska et al. [47]. In this work, authors obtained thermally stable and transparent non-crystalline oxyfluoride glasses with high ΔT factors for low 4–12 mol% of P_2_O_5_ concentration. The stability parameter ΔT of the mentioned fluoroindate glasses increased from 114 to 155 °C with an increase in P_2_O_5_ concentration [47].

The effect of erbium doping concentration on the thermal properties of tellurite glasses was studied in the literature. These data relate mainly to oxide glasses and report that the increasing amount of erbium ions increases the characteristic temperatures and thermal stability of tellurite glasses [48,49,50]. The transition temperature of the TZNE (TeO_2_-ZnO-Na_2_O-Er_2_O_3_) glass increased by 13 °C with the increasing Er_2_O_3_ content from 0.5 to 2.5 mol% [48]. According to the literature [49], doping the TZN (TeO_2_-ZnO-Na_2_O) glass up to 2 mol% of Er_2_O_3_ showed the increase of the transition temperature T_g_ and the thermal stability ΔT. For the above-mentioned concentration of erbium ions, the authors noted a decrease in the thermal stability of the analyzed glass [49]. However, the introduction of Er_2_O_3_ from 0.5 to 2.8 mol% resulted in increase of the T_g_ value and the thermal stability of the TeO_2_-Li_2_O-ZnO-Nb_2_O_5_-Er_2_O_3_ glass (TLZNE) [50]. 

In order to analyze the effect of ErF_3_ concentration on the thermal properties of the precursor glass (TP glass), DSC curves of oxyfluoride erbium-doped phospho-tellurite glass samples were measured and shown in Figure 2. Based on obtained DSC curves the characteristic temperatures such as the glass transition T_g_ (onset), the crystallization T_c_ (in the maximum), and the melting T_m_ temperatures were obtained. Based on these parameters the stability factor ΔT was calculated. The results of thermal parameters for different ErF_3_ concentrations into the chemical composition of the TP glass are presented in Table 1. It can be observed that the glass transition temperature T_g_ increases from 313 ± 1 to 329 ± 1 °C with the increase of ErF_3_ concentration into the chemical composition of precursor glass. The same trend was observed in the literature [51]. Jha et al. related the increase of transition temperature T_g_ by the increase of ErF_3_ to the strong bonding of erbium ions with non-bringing oxygens, which led to the increase of rigidity of the glass network [51]. 

All DSC curves showed the existence of exothermic peak/peaks in the maximum in the 430–485 °C range (Figure 2, Table 1), which indicates that partial crystallization has taken place in the oxyfluoride erbium-doped phospho-tellurite glasses. In the DCS curve of glass doping with 0.25 mol% of ErF_3_ (TP0.25ErF_3_ glass)_,_ the maximum of crystallization peak is located at around 438 ± 1 °C. With the increase of the erbium fluoride concentration to 0.5 mol%, the position of mentioned peak shifted to a higher temperature (458 ± 1 °C) and additionally, the second crystallization peak appeared (at 415 ± 1 °C) – glass TP0.50ErF_3_. The exothermic peak in the DSC curve of TP0.75ErF_3_ glass split into two peaks, at 451 ± 1 and 483 ± 1 °C. A wide exothermic peak is visible in the DSC thermal curve of TP1.00ErF_3_ glass. Related to the DSC curve of TP0.75ErF_3_ glass, it can be deduced that two exothermic peaks overlapped, therefore the crystallization temperature of TP1.00ErF_3_ glass has been determined to the maximum of exothermic effect and is 466 ± 1 °C.

For 1.25 mol% of ErF_3_ glass, two crystallization peaks were observed in the maximum at 420 ± 1 and 456 ± 1 °C, respectively. It is worth noting that the crystallization peak in the DSC curves of TP0.75ErF_3_, TP1.00ErF_3_, and TP1.25ErF_3_ glasses has a greater value of full width at half maximum (FWHM) compared to the FWHM of the exothermic peak at 458 ± 1 °C (TP0.50ErF_3_) and at 438 ± 1 °C (TP0.25ErF_3_). The melting temperature T_m_ seems to be not affected by the increase of ErF_3_ (Figure 2 and Table 1).

As can be seen in Table 1, oxyfluoride phospho-tellurite TP0.25ErF_3_, TP0.75ErF_3_, TP1.00ErF_3_, and TP1.25ErF_3_ glasses have a thermal stability > 100 °C (ΔT = 109–142 ± 1 °C), except TP0.50ErF_3_ glass, which has a smaller value < 100 °C (ΔT = 95 ± 1 °C). Therefore, TP0.25ErF_3_, TP0.75ErF_3_, TP1.00ErF_3_, TP1.25ErF_3_ glasses should be suitable for optical fiber drawing [52,53,54]. In comparison with other oxyfluoride tellurite glasses, the value of ΔT (case of 1 mol% of ErF_3_) is larger than that of TeO_2_-ZnF_2_-ZnO-Er_2_O_3_ glasses (ΔT = 98–126 °C) [55], TeO_2_-ZnO-La_2_O_3_-Tm_2_O_3_-Yb_2_O_3_ (ΔT = 126–135 °C) [56], TeO_2_-GeO_2_-InO_2/3_-ZnO-KF (ΔT = 129 °C) [57], and TeO_2_-ZnO-7ZnF_2_ (ΔT = 120 °C) [51].

#### 3.1.3. Structural Studies of Oxyfluoride Erbium-Doped Phospho-Tellurite Glass

In order to investigate the evolution of erbium-doped phospho-tellurite glass structure, FTIR and Raman measurement were carried out on the precursor glass TP0.25ErF_3_ doped with varying concentrations of ErF_3_.

Figure 3 shows the normalized to the band at ~1030 cm^−1^ FTIR spectra (in the 1300–500 cm^−1^ range) of the precursor oxyfluoride phospho-tellurite glass with varying ErF_3_ concentrations. As shown in Figure 3, all spectra presented the typical bands characteristic for tellurite glasses, i.e., at 610, 710, 770 cm^−1^ [58,59,60] and phosphate glasses, i.e., the bands at 500 cm^−1^ and in the 1300–900 cm^−1^ range [61,62,63,64,65,66,67]. 

On the basis of Figure 3 it could be concluded that the increase in ErF_3_ does not result in the incorporation of erbium ions in the glass network, because all FTIR spectra are very similar. Most likely, erbium ions act as the network modifier [51]. In order to better understand the influence of erbium trifluoride addition on the oxyfluoride phospho-tellurite glass structure, the deconvolution of the FTIR spectra of TP0.25ErF_3_, TP0.75ErF_3_, and TP1.25ErF_3_ samples were carried out. Figure 4, Figure 5 and Figure 6 present a deconvolution of the FTIR spectra of samples with the selected value of ErF_3_. Figure 7 presents a plot of bands position and their integral intensity (area under the curve in %) for deconvoluted TP0.25ErF_3_, TP0.75ErF_3_, and TP1.25ErF_3_ spectra. The parameters of deconvoluted spectra and bands assignment to the appropriate vibrations are shown in Table 2 and Table 3, respectively [68,69,70,71,72,73,74,75,76,77]. 

Bands A at ~545 cm^−1^ and B at ~566 cm^−1^ correspond to the deformation vibration of δO–P–O and δO–P–O bonds in Q^2^ units [63]. The bands C and D at ~606 and ~680 cm^−1^ are due to the stretching vibrations of Te–O bonds in trigonal bipyramidal units TeO_4_ (tbp) [69,70]. B and E at ~740 cm^−1^ relates to the stretching vibrations of trigonal pyramidal units TeO_3_ (tp) or TeO_3+1_ polyhedra [70]. B and F at ~780 cm^−1^ is assigned to the vibration of the continuous network composed of TeO_4_ and Te–O stretching vibration of TeO_3+1_ polyhedra [69] or symmetric P–O=P bonds in Q^1^ units [71]. B and G at ~800 cm^−1^ is ascribed to the asymmetric stretching vibrations of TeO_3_ (tp) units or TeO_3+1_ polyhedra [69]. 

In the FTIR spectra of phosphate glasses, there are bands associated with Q^3^–Q^0^ groups (where Q is the number of bridging bonds) [72,73,74,75,76,77]. B and H at ~870 cm^−1^ relates to the asymmetric stretching vibrations of Q^1^ units [72,73]. B and I at ~920 cm^−1^ is ascribed to the stretching vibrations of P–O–P linked with metaphosphate chain and P–F groups in Q^2^ units [74,75]. The bands at ~970, ~1020, ~1077/1094/1102 and ~1135/1158/1178 cm^−1^ are due to the asymmetric stretching vibration of P–O^−^ bonds in Q^0^ units [68], the symmetric stretching vibrations of PO_3_ groups in Q^1^ units [74], the asymmetric stretching vibrations of PO^2−^_3_ groups in Q^1^ units [76], and the asymmetric stretching vibrations of non-bridging oxygen in Q^2^ units [77], respectively. 

As can be seen in Figure 4, Figure 5, Figure 6 and Figure 7, Table 2 and Table 3, the sum of the bands intensity at ~1020 and ~1070–1110 cm^−1^ (both related to Q^1^ units) increased from 27% to 41%, while the integral intensity of the bands at ~970 cm^−1^ (related to Q^0^ units) and at 1130–1180 cm^−1^ (related to Q^2^ units) decreased (from 10% to 4% and 15% to 4%, respectively) with increasing ErF_3_ concentration. This clearly indicates that the addition of ErF_3_ results in an increase of Q^1^ units at the expense of Q^0^ and Q^2^ units, according to the following equation: Q^2^ + Q^0^ ↔ 2Q^1^ [78]. The P–(O, F)–P bonds are broken and the Er–(O, F)–P (NBO, non-bridging oxygen) bonds are formed as the number of erbium ions increased, which means that the phosphate chain is becoming shorter [79,80].

As seen in Figure 4, Figure 5, Figure 6 and Figure 7, the position of the bands at ~1077 and ~1135 cm^−1^ shifted to the higher wavenumbers with the increasing ErF_3_ content in the glass. This might be due to the replacement of fluoride for oxide ions leading to the increase in the bond strength. This confirms the formation of F–P–F bonds [23]. 

The cut-off frequency of vibrational modes, related to the maximum phonon energy of the analyzed glass network, occurs at ~1180 cm^−1^ for 1.25 mol% ErF_3_ concentration (Figure 6). This value is lower than found in the literature for phospho-tellurite glasses [43,44,61].

Figure 8 shows normalized Raman spectra of glasses doped with various erbium trifluoride content in the 1300–500 cm^−1^ range. The Raman spectra of all samples (Figure 8) have bands that are attributed to the bond vibration occurring in the tellurium (300–800 cm^−1^) [81,82,83,84,85] as well as phosphate (900–1300 cm^−1^) [86,87,88,89,90] glass network. Additionally for Raman spectra, it was necessary to deconvolution selected spectra for better understanding the structure of glasses (Figure 9, Figure 10 and Figure 11). Figure 12 presents a plot of Raman bands position and their integral intensity for deconvoluted TP0.25ErF_3_, TP0.75ErF_3_ and TP1.25ErF_3_ spectra. The parameters of deconvoluted Raman spectra and bands assignment to appropriate vibrations are shown in Table 4 and Table 5, respectively.

According to the literature data, the Raman bands in the 360–580 cm^−1^ range can be attributed to the bending vibration of Te–(O, F)–Te or O,F–Te–O,F bands of [Te(O, F)_4_] trigonal bipyramidal units [81,82]. The band at around 669/653/645 cm^−1^can be assigned to the stretching variation of Te–O, F bonds in [Te(O, F)_4_] units [83]. The band at around 714–704 cm^−1^is assigned to the Te(O, F)_4_ tbp units [84]. The band located at 780 cm^−1^ can be ascribed to Te–O^−^ stretching vibration in [TeO_3_] trigonal pyramids or symmetric stretching vibration in [TeO_3+1_] units [85]. 

As reported in the literature [87], the band at around 870–880 cm^−1^ is due to symmetric stretching vibration of the P–F bonds. As can be seen in Figure 9, Figure 10, Figure 11 and Figure 12 and in Table 4 and Table 5, the integral intensity of this band increased with the addition of erbium trifluoride. This may be due to the replacement of fluorine for oxygen ions leading to the increase in bond strength [88]. 

All Raman spectra of glasses modified with various ErF_3_ content do not have bands above 1300 cm^−1^ (Figure 9), the presence of which is attributed to the Q^3^ units. This means that there are no Q^3^ units in the structure of the analyzed glasses [89].

In all deconvoluted Raman spectra of TeP0.25ErF_3_, TeP0.75ErF_3_, and TeP1.25ErF_3_ glasses there is a band at about 950 cm^−1^, which is attributed to symmetric PO_4_ stretch on Q^0^ tetrahedra [25,90]. A decrease in the intensity of this band from 16% ± 3% (TeP0.25ErF_3_ glass) to 5% ± 3% (TeP1.25ErF_3_ glass) indicates a reduction in the number of Q^0^ units as the ErF_3_ content increases (Table 4 and Table 5). The band at about 1020–1040 cm^−1^ (Figure 9, Figure 10, Figure 11 and Figure 12) can be attributed to the stretching vibrations of P-O bridging bonds in Q^1^ units [89]. The integral intensity of this band increased from 2% ± 1% (TeP0.25ErF_3_ glass) to 5% ± 1% (TeP0.25ErF_3_ glass) (Table 4 and Table 5). This indicates an increase in Q^1^ units with the addition of erbium trifluoride. As can be seen in Figure 9, Figure 10, Figure 11 and Figure 12, with the increasing amount of ErF_3_ there was a decrease in the intensity of the integral band at about 1090–1150 cm^−1^ (from 3% ± 0.5% for TeP0.25ErF_3_ glass to 1% ± 0.5% for TeP1.25ErF_3_ glass), attributed to symmetrical stretching vibrations of non-bridging PO_2_ bonds of Q^2^ units [23].

In summary, it can be concluded that with the increase in the content of Er^3+^ and F^−^ ions in the structure of the glasses, the integral intensity of the bands associated with Q^0^ (band at ~950 cm^−1^) and Q^2^ (band at ~1090–1150 cm^−1^) units decreased at the expense of an increase in the integral intensity of the band associated with Q^1^ units (band at ~1020–1040 cm^−1^)—Figure 8, Figure 9, Figure 10, Figure 11 and Figure 12, Table 4 and Table 5. This confirms the conclusions from the interpretation of the FTIR spectra. Fluoride ions are embedded in the glass structure (F^−^ replace O^−^ anions), while erbium ions depolymerized the phosphate network and phosphate chain is becoming shorter [23,25,89,90]. 

#### 3.1.4. Spectroscopic Properties of Oxyfluoride Phospho-Tellurite Glass Doped Er^3+^

In general, the luminescent properties of RE ions in the inorganic glasses depend on the chemical composition of the host glass, the activator concentration (lanthanide ions as optically active dopants), and the excitation power (pumping system). These factors and in particular the oxyfluoride environment (phonon energy) of the active ion significantly affect the emission efficiency of excited states of Er^3+^ ions in the glasses [91,92,93,94]. It is also discussed in our investigated system glass and GC TP system.

MIR (mid-infrared) emission spectra of oxyfluoride Er^3+^-doped phospho-tellurite glasses in the 2400–2900 nm range recorded under 976 nm LD excitation at room temperature, is shown in Figure 13. The emission peak at 2725 nm corresponding to ^4^I_11/2_→^4^I_13/2_ transition is observed. The changes in the intensity of this peak are characterized for all concentrations of erbium ions in oxyfluoride phosphate-tellurite glass (inset of Figure 13). The 2725 nm emission intensity increased with the increase of ErF_3_ concentration (Figure 13). A similar effect was observed in tellurite glass [95].

The luminescence spectra of Er^3+^-doped phospho-tellurite glass recorded at room temperature in the wavelength of 1400–1700 nm under the excitation of 976 nm, is shown in Figure 14. The NIR intensity of emission line at 1532 nm occurred by ^4^I_13/2_→^4^I_15/2_ transition increased with the increase of ErF_3_ concentration up to 1.25 mol% in precursor glass—the inset of Figure 14.

Figure 15 shows the emission spectra of erbium-doped phospho-tellurite glasses obtained in the visible regions under excitation at 976 nm. The observed emission bands are attributed to ^2^H_11/2_→^4^I_15/2_ (528 nm), ^4^S_3/2_→^4^I_15/2_ (551 nm), and weak emission band at 668 nm to ^4^F_9/2_→^4^I_15/2_ transitions of Er^3+^, respectively. As can be seen in Figure 15, glasses have a strong green emission at 528 and 551 nm and a weak red emission at 668 nm. The green upconversion emission corresponding to ^2^H_11/2_,^4^S_3/2_→^4^I_15/2_ transition is dominant. This phenomenon can be explained as follows: Two dominant mechanisms are involved in the upconversion process: (1) the excited state absorption (ESA) and (2) energy transfer upconversion (ETU) (Figure 16). In the case of low doping level (0.25 mol% of ErF_3_ in the precursor glass), erbium ions are scarcely distributed in the glassy phase. The ^2^H_11/2_ and ^4^S_3/2_ levels, corresponding to the green emission at 525 and 545 nm, are populated by the multiphonon relaxation from the ^4^F_7/2_ level, thus the 525 nm emission intensity is reduced and stronger emission signals at 545 nm than that at 525 nm were obtained. For high Er^3+^ concentration (more than 0.5 mol% of the ErF_3_) is greatly promoting the ETU1 process of excited Er^3+^ (^4^I_11/2_) from one to another and ultimately excited to ^4^F_7/2_ level. Similar behavior was observed in oxyfluoride glass-ceramics containing LaF_3_ nanocrystals [96].

The inset of Figure 15 shows the relation between the UC luminescence intensity I_UP_ as a function of the pumping power I_IR_. It is well known that the relation expressed as ${I\;}_{\mathrm{UP}}\propto {\;I}_{\mathrm{IR}}^{m}$, where *m* determines the number of photons of the optical pump used in the conversion of excitation process which occurs in a given structure of the energy levels of RE elements [97].

The mechanism of the upconversion process was determined from the log-log dependence of the emission intensity on the excitation power and presented for the precursor glass doped with the highest content of erbium trifluoride (Figure 17). The slopes of 1.32 and 1.24 for green and red transitions of erbium ions indicates that the 2-photon mechanism was involved in the upconversion luminescence process (Figure 17) [96,98]. 

### 3.2. Studies of Glass-Ceramic 

#### 3.2.1. X-ray Diffraction Analysis of Transparent Glass-Ceramic

The TP1.25ErF_3_ glass was heat-treated to induce crystallization and form glass-ceramic. To get a transparent glass-ceramic, the heat treatment temperature was selected to be 400 °C. The sample of TP1.25ErF_3_ glass was heat-treated at 400 °C for 3 h. The X-ray diffraction pattern of TP1.25ErF_3_ glass heat-treated at 400 °C is presented in Figure 18. As shown in Figure 1, the precursor phospho-tellurite glass doped with 1.25 mol% of ErF_3_ is amorphous with no diffraction peaks. After the heat-treated process the diffraction peaks are clearly observed, next to the broad and intense broad hump in the 20–35° range of two theta angle (Figure 18). These peaks are assigned in 100% to barium difluoride (BaF_2_) cubic phase (JCPDF: 00-001-0533). The size of precipitated crystals in the obtained glass-ceramic was calculated on the basis of Scherrer‘s formula and evaluated to be about 16 nm [99]. The value of the calculated lattice parameter of BaF_2_ crystals is 0.5899 ± 0.0002 nm and was smaller than the JCPDF value of 0.6187 nm. The similar shrinkage of barium difluoride crystals lattice was observed in the SiO_2_-ZnF_2_-BaF_2_-ErF_3_-YbF_3_ glass [99] and can be ascribed to the entrance of erbium ions into BaF_2_ nanocrystals because the radius of erbium ions is smaller than that of barium ions [100,101,102].

#### 3.2.2. Spectroscopic Properties of the Transparent Glass-Ceramic

Rare-earth doped oxyfluoride glass-ceramics have research interest for attracted combined advantages of low phonon energy of fluoride and very good mechanical and chemical properties of the oxide matrix [103,104]. In the literature many investigations can be found, which have been done on the rare earth ions doped glass-ceramic containing lead or cadmium fluoride nanocrystals [105,106,107,108]. However, cadmium fluoride CdF_2_ and lead fluoride PbF_2_ are poisonous raw materials, which limited their application. This is reason that the rare earth doped transparent glass-ceramic containing nanocrystals without the toxic ingredients such as Pb or Cd is still evolving [109,110,111,112,113]. Much effort has been devoted to the search for novel transparent glass-ceramic. The glass-ceramic with LaF_3_ nanocrystals was reported for the first time by Dejneka as non-toxic glass-ceramic [114]. Next, researchers studied systemically the preparation of the rare-earth-doped glass-ceramic containing lanthanide trifluoride nanocrystals [115,116,117,118,119,120,121,122,123,124]. 

The rare-earth-doped fluorides such as CaF_2_, BaF_2_, or SrF_2_ in the various glass host have been extensively investigated [125,126,127,128,129,130]. These alkaline-earth fluorides have the same fluorite structure as the β-PbF_2_. In the alkaline-earth fluorides glass-ceramic, the divalent alkaline-earths cations (Ca^2+^, Ba^2+^, or Sr^2+^) may be substituted by trivalent rare-earth cations [131]. 

The optical properties of BaF_2_ make it a universal optical material. Optical properties of BaF_2_ are related to the structural and electronic properties of barium difluoride such as very large bandgap (11 eV), the low phonon energy (319 cm^−1^) and the large optical transparency from ultraviolet (UV) to far-infrared (FIR) from 0.2 to 14.0 μm [132,133]. 

In the research, it may be found that rare-earth ions like Er^3+^ are the luminescence centers in BaF_2_ nanocrystals in the glass-ceramic [101,134,135,136]. Chen et al. obtained transparent glass-ceramics containing BaF_2_ nanocrystals doped with Er^3+^, prepared by the sol-gel route. The authors recorded the efficient upconversion emissions around 545, 565, and 655 nm under 980 nm excitation due to the lower phonon energy environment of Er^3+^ ions in glass-ceramic [85]. Transparent glass-ceramic with erbium-doped BaF_2_ nanocrystals in the fluoroborate system was synthesized for Shinozaki et al. [134]. In this paper, the authors presented that crystallization enhanced the luminescence intensity 30 times compared to the precursor glass [134]. The enhanced upconversion luminescence of the Er^3+^ ions in transparent oxyfluoride GCs containing BaF_2_ nanocrystals in the SiO_2_-ZnF_2_-BaF_2_-ErF_3_ system has been also investigated by Qiao et al. [135]. The erbium-doped germano-gallate oxyfluoride glass-ceramics containing BaF_2_ nanocrystals with the high transmittance in the mid-infrared region were prepared by Zhao et al. [136]. Unfortunately, to the best of our knowledge, no reports of the transparent Er^3+^-doped BaF_2_ glass-ceramic in the oxyfluoride phospho-tellurite exist in the literature. An issue which this report is intended to address.

Figure 19 shows the MIR emission spectra of the TP1.25ErF_3_ glass and glass-ceramic under 980 nm LD excitation recorded at room temperature. Enhanced emission at a wavelength of 2725 nm was observed in glass-ceramic. The influence of the heat-treatment process on the intensity of MIR luminescence shows that for 3 h of the annealing of the TP1.25ErF_3_ glass sample, the 30% enhancement was observed for the glass-ceramic (Figure 19). 

The higher NIR luminescence intensity exhibited (22% enhancement) when the TP1.25ErF_3_ glass sample was annealed for 3 h (Figure 20). 

In the case of visible emission of Er^3+^ ions, the glass-ceramic was characterized by the higher intensity at 551, 528, and 668 nm (Figure 21) in comparison to the TP1.25ErF_3_ glass (Figure 16). It is worth noting that the shape of UC luminescence in the green bands is narrower for the GC sample than the parent glass. Thus, it is confirmed that the structural changes in the vicinity of erbium ions. 

The intensity of MIR, NIR, and upconversion emission spectra of glass-ceramic were stronger compared to the intensity emission spectra of the TP1.25ErF_3_ glass sample. This is most likely due to incorporated erbium ions in the crystalline environment of BaF_2_ nanocrystals [129,130,131].

## 4. Discussion 

Rare-earth doped oxyfluoride glass-ceramics have been extensively investigated due to potential applications, i.e., in solid laser and fiber amplifiers. Oxyfluoride glass-ceramics are more appropriate for practical applications compared to oxide and fluoride glasses, because the glass-ceramics have the lower phonon energy than the oxide glasses and excellent chemical durability and mechanical strength compared to the fluoride glasses. 

Considering the above, authors of this paper first investigated precursor oxyfluoride phospho-tellurite glass in the TeO_2_-P_2_O_5_-BaF_2_-ZnF_2_-Na_2_O system. Addition of the second glass former (P_2_O_5_) was aimed at improving the thermal stability and the tensile strength of tellurite glass. Next, oxyfluoride phospho-tellurite precursor glass was doped with erbium trifluoride (ErF_3_). In this system, thermal and chemical stable glass with 0.25, 0.50, 0.75. 1.00, and 1.25 mol% of ErF_3_ was obtained. Above 1.25 mol% of ErF_3_ (for 1.5 mol% of ErF_3_), we obtained non-transparent material. This showed that the agglomeration of Er^3+^ ions has occurred. 

The thermal, structural, and optical studies were performed for precursor TeO_2_-P_2_O_5_-BaF_2_-ZnF_2_-Na_2_O glass with varying ErF_3_ concentrations (up to 1.25 mol%). The thermal characteristic of erbium-doped oxyfluoride phospho-tellurite glass provided the information of characteristic temperatures. The glass transition temperature T_g_ increased from 313 to 329 °C with the increase of ErF_3_ concentration into the chemical composition of precursor glass. This increase in the transition temperature indicates a more strongly bound network in the glass doped with ErF_3_. A stronger ionic cross-linking between modifier cations (Er^3+^) and non-bridging oxygens (NBO) has taken place, which led to the increase of rigidity of the glass network. The existence of clear exothermic peak/peaks in the 430–485 °C range indicated that crystallization occurred in the erbium-doped oxyfluoride phospho-tellurite glasses. The crystalline phase, obtained after 1 h annealing in the maximum of each crystallization peak was only cubic barium difluoride (BaF_2_). These data were used to synthesis and characterization of transparent glass-ceramics with barium difluoride nanocrystals. Moreover, the oxyfluoride phospho-tellurite glass with various concentration of erbium fluoride can be considered as a good thermal stability. These glasses should be suitable for optical fiber drawing.

FTIR and Raman spectra analysis confirmed the presence of tellurite and phosphate units in the precursor glass doped with ErF_3_. The replacement of fluoride for oxide leading to an increase in the bond strength confirms the formation of F–P–F bonds when TeO_2_ is replaced by ErF_3._ Comparison of deconvoluted spectra of TP0.25ErF_3_, TP0.75ErF_3_, and TP1.25ErF_3_ glasses indicated that the erbium ions were modifier ions and depolymerized phosphate network. It was found, that the addition of ErF_3_ resulted in the increase of Q^1^ phosphate units at the expense of Q^0^ and Q^2^ phosphate units. The P–(O, F)–P bonds were broken and the Er–(O, F)–P (NBO, non-bridging oxygen) bonds were formed as the amount of ErF_3_ increased up to 1.25 mol%. The phosphate chains were becoming shorter. A possible mechanism that explains the structural modifications by Er^3+^ in the erbium-doped oxyfluoride phospho-tellurite glass can be related to the electronegativity of the structural units in a network of glass. According to Rada and Culea [137] the presence of multiple oxides/fluorides in the glass increases the tendency of network formers to attract oxygen/fluoride ions. This is due to the competition between the cations themselves and this preference. The units, which have higher electronegativity value pick up oxygen or fluoride ions and get modifiers. For erbium trifluoride, the erbium ions are firstly inserted in the trivalent state and they are considered as the modifiers ions due to the strong affinity of Er^3+^ toward groups containing negative charged non-bridging oxygens. Both tellurium and phosphorus cations in the glass matrix attract fluoride anions, which yield a competition between them. Since phosphorus pentoxide has a higher electronegativity value than tellurium dioxide, P_2_O_5_ picked up fluoride ion. Hereby, for 1.25 mol% of ErF_3_, erbium ions participated in the glass network as the modifier ions for Q^2^ phosphate units [137]. 

The spectroscopic study showed, that the emission peak at around 2725 nm corresponding to ^4^I_11/2_→^4^I_13/2_ transition was observed in the mid-infrared emission spectra of all doped glasses. The near-infrared emission line at 1533 nm occurred by ^4^I_13/2_→^4^I_15/2_ transition. In the emission spectra in the visible regions of erbium-doped phospho-tellurite glasses bands, attributed to ^2^H_11/2_→^4^I_15/2_ (528 nm), ^4^S_3/2_→^4^I_15/2_ (551 nm), and ^4^F_9/2_→^4^I_15/2_ (668 nm) transitions of Er^3+^ were observed, respectively. The glasses have a strong green emission at 528 and 551 nm and a weak red emission at 668 nm. The green upconversion emission corresponding to ^2^H_11/2_,^4^S_3/2_→^4^I_15/2_ transition is dominant. The mechanism of the upconversion process was determined from the log-log dependence of emission intensity on the excitation power and indicated that the 2-photon mechanism was involved in the upconversion luminescence process. Finally, the intensity of mid-, near-infrared emission spectra and upconversion of erbium-doped oxyfluoride glass exhibits an increased tendency with the increment of Er^3+^ concentration until 1.25 mol%. The intensity of the emission peaks has been enhanced with increasing the mol% ErF_3_ in the precursor glass. It suggests that the observed spectra arisen due to the presence of the erbium trifluoride in the glass and proportional to its concentration.

By annealing of the precursor oxyfluoride glass doping with the highest value of ErF_3_ (1.25 mol%) for 3 h in 400 °C we obtained oxyfluoride transparent phospho-tellurite glass-ceramic with Er^3+^-doped barium difluoride nanocrystals with the size of 16 nm. Compared to the literature [55,138,139], in our transparent glass-ceramic, single nanocrystals (100% of BaF_2_) appeared after the heat treatment. 

To check if the Er^3+^ ions are incorporated into the BaF_2_ nanocrystals, the mid- and near-infrared emission spectra of the glass-ceramics and upconversion were measured under excitation at 976 nm in room temperature. The glass-ceramic sample has a stronger 2725 nm emission intensity than the base glass. The 30% enhancement of the intensity band at 2725 nm was observed for glass-ceramic. The near-infrared emission of the glass-ceramic that corresponds to Er^3+^: ^4^I_13/2_ → ^4^I_15/2_ was 22% enhanced compared to the glass. Under 980 nm LD pumping, the green upconversion intensity of Er^3+^ in the glass-ceramic was observed much stronger than that in the glass. The cubic site of Er^3+^ and a low vibration frequency of its BaF_2_ environment resulted in the enhancement of spectroscopic properties of the glass-ceramic compared to the oxyfluoride phospho-tellurite glass with 1.25 mol% of ErF_3_. 

The data presented in this paper first report of the transparent Er^3+^-doped BaF_2_ glass-ceramic in the oxyfluoride phospho-tellurite system. The following step will be the investigation of luminescence kinetic of obtained glass-ceramic. It might provide useful information for further development of this material in both RE optimization and composition modifications leading to photonics applications. 

## 5. Conclusions

In summary, Er^3+^-doped oxyfluoride phospho-tellurite glasses were prepared in the 10P_2_O_5_-45(BaF_2_-ZnF_2_)-5Na_2_O-xErF_3_ system by adding x = 0.25, 0.50, 0.75, 1.00, and 1.25 mol% of ErF_3_ and characterized for their thermal, structure, and spectroscopic properties. An increase in the glass transition temperature of glasses with the addition of increase of ErF_3_ into the chemical composition of precursor glass indicated a stronger ionic cross-linking between Er^3+^ and non-bridging oxygens (NBO) atom. FTIR and Raman analysis confirmed that the addition of ErF_3_ resulted in an increase of Q^1^ at the expense of Q^0^ and Q^2^ phosphate units. The erbium ions were modifier ions and depolymerized phosphate network. The intensity of the mid-, near-infrared and upconversion emission peaks showed the increase with the increasing mol% of ErF_3_ in the precursor glass. Transparent oxyfluoride phospho-tellurite glass-ceramics containing cubic BaF_2_ nanocrystals were synthesized by heat treatment of the base glass containing 1.25 mol% ErF_3_. The X-ray diffraction (XRD) analysis revealed the formation of 16 nm barium difluoride nanocrystals in the oxyfluoride phospho-tellurite glass-ceramics. Preferential incorporation of erbium ions into the BaF_2_ nanocrystals was confirmed by the infrared and upconversion emission spectra. The low vibration frequency of erbium-doped BaF_2_ nanocrystals resulted in the enhancement of spectroscopic properties of the phospho-tellurite glass-ceramic compared to the base glass. 

## Figures and Tables

**Figure 1 materials-12-03429-f001:**
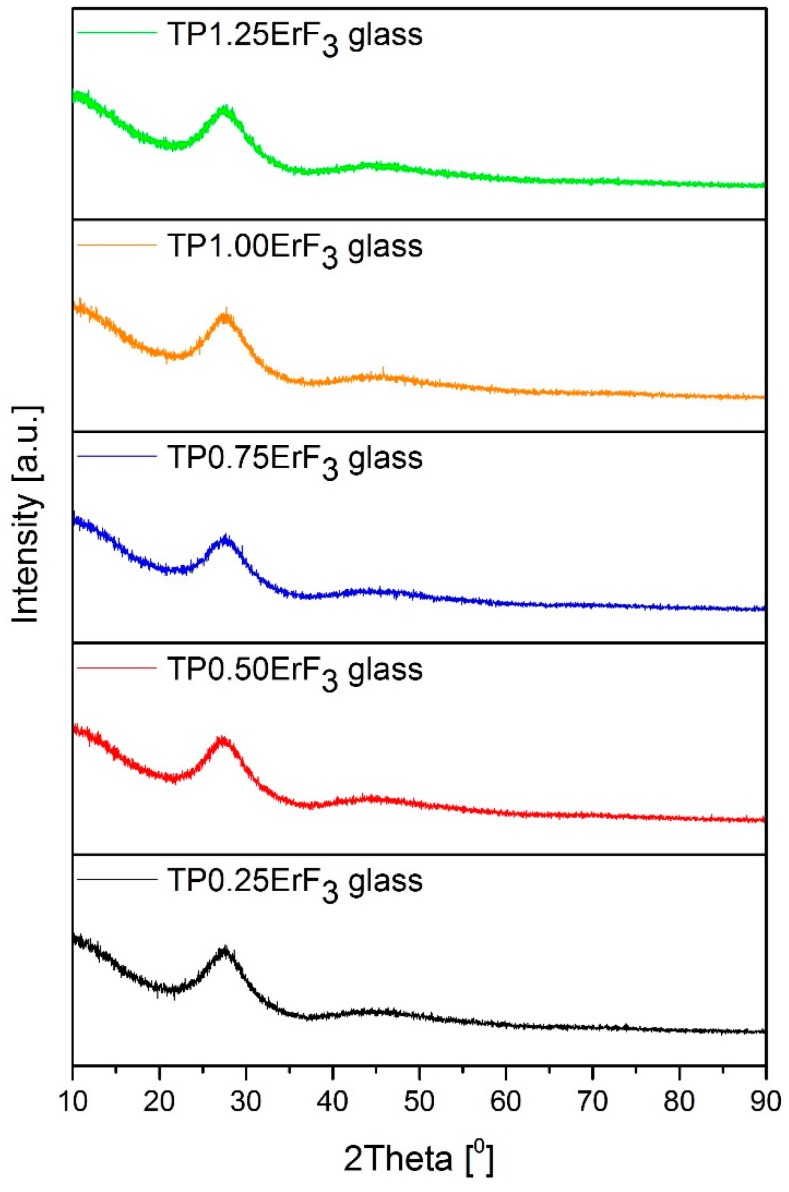
X-ray diffraction patterns of oxyfluoride phospho-tellurite glass doped with erbium trifluoride (ErF_3_).

**Figure 2 materials-12-03429-f002:**
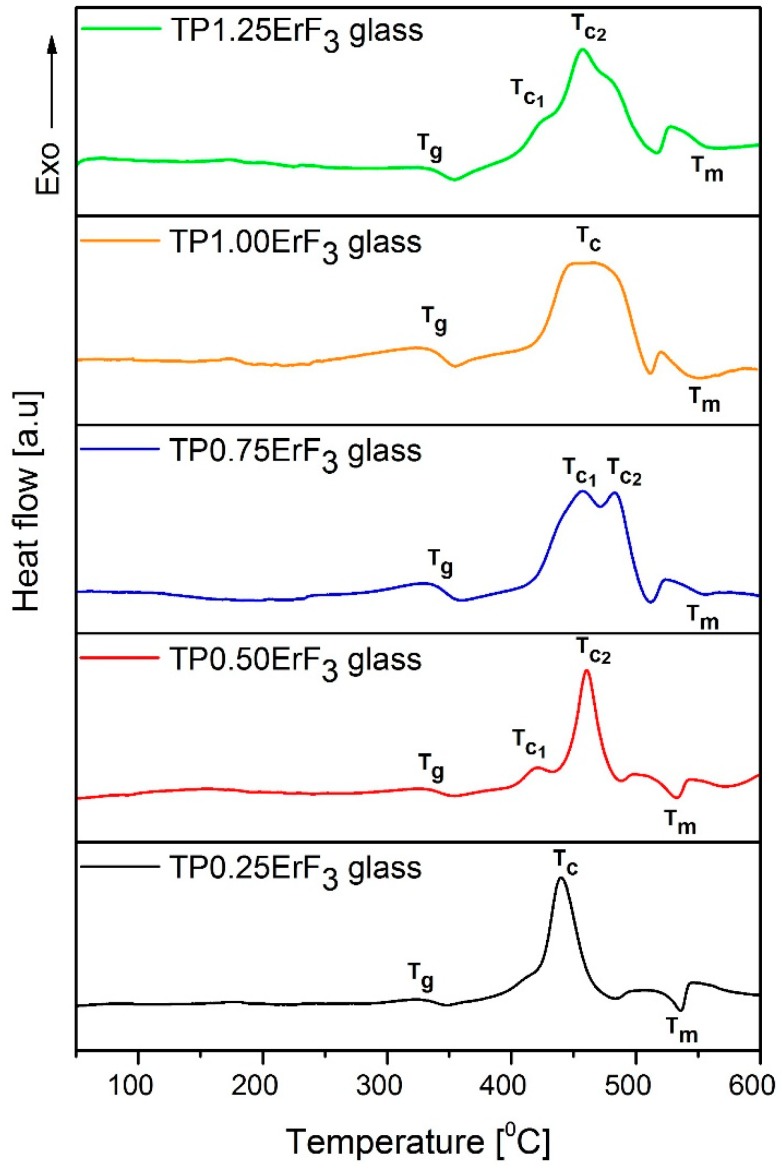
DSC curves of oxyfluoride phospho-tellurite glass doped with ErF_3_.

**Figure 3 materials-12-03429-f003:**
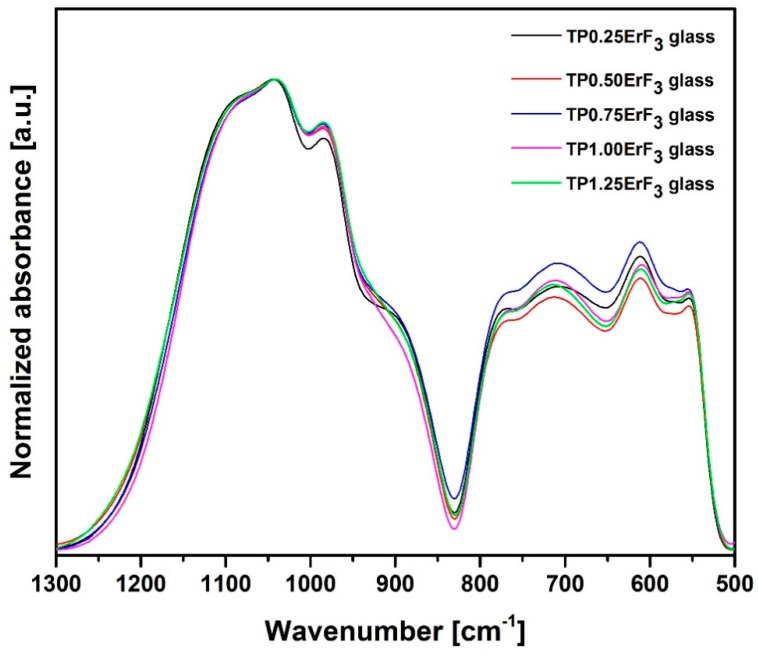
FTIR spectra of precursor glass doped with varying ErF_3_ content.

**Figure 4 materials-12-03429-f004:**
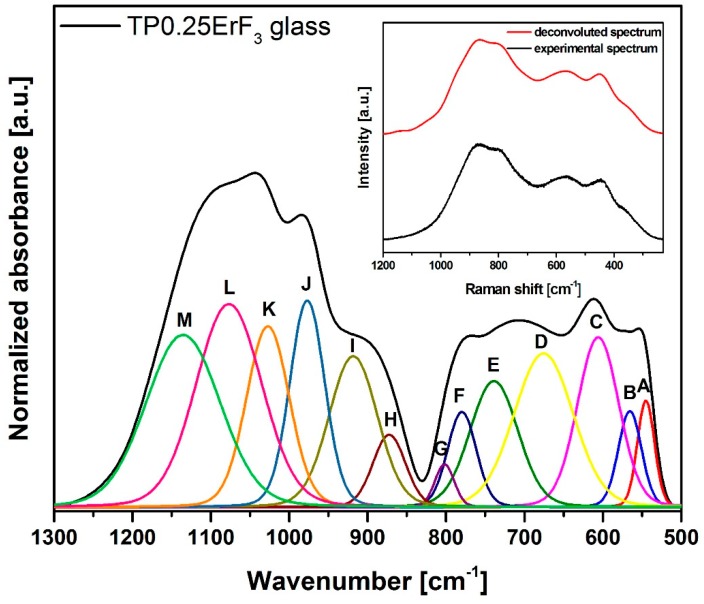
Deconvoluted FTIR spectra of TP0.25ErF_3_ glass. (inset) Deconvoluted and experimental spectra.

**Figure 5 materials-12-03429-f005:**
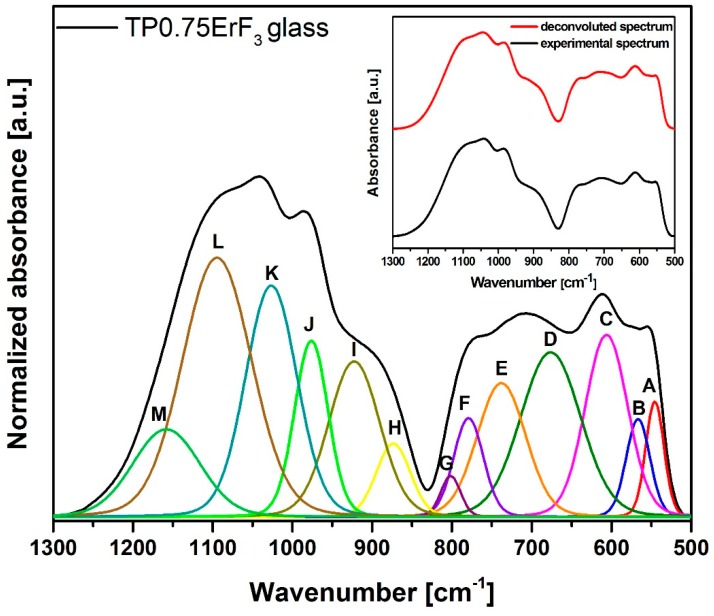
Deconvoluted FTIR spectra of TP0.75ErF_3_ glass. (inset) Deconvoluted and experimental spectra.

**Figure 6 materials-12-03429-f006:**
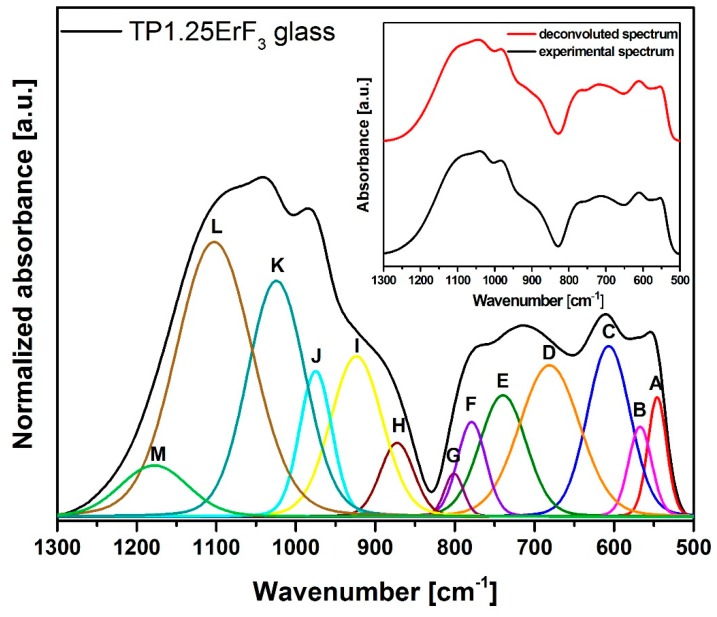
Deconvoluted FTIR spectra of TP1.25ErF_3_ glass. (inset) Deconvoluted and experimental spectra.

**Figure 7 materials-12-03429-f007:**
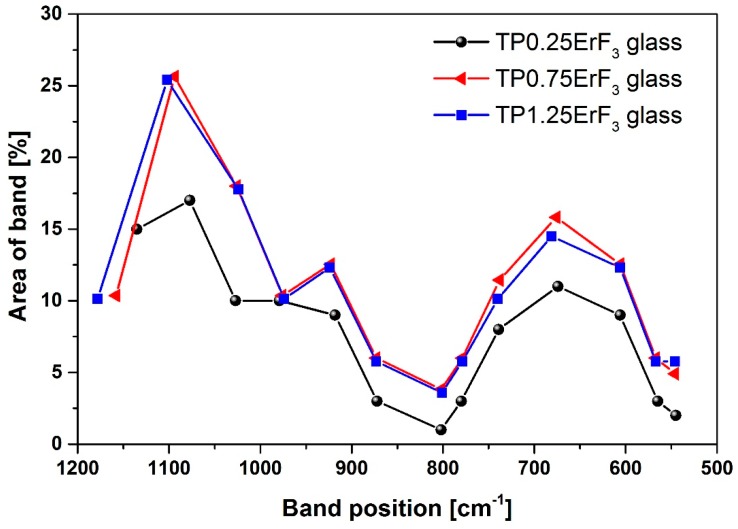
Plot of bands position and their area for deconvoluted FTIR spectra of the TP0.25ErF_3_, TP0.75ErF_3_, and TP1.25ErF_3_ glasses.

**Figure 8 materials-12-03429-f008:**
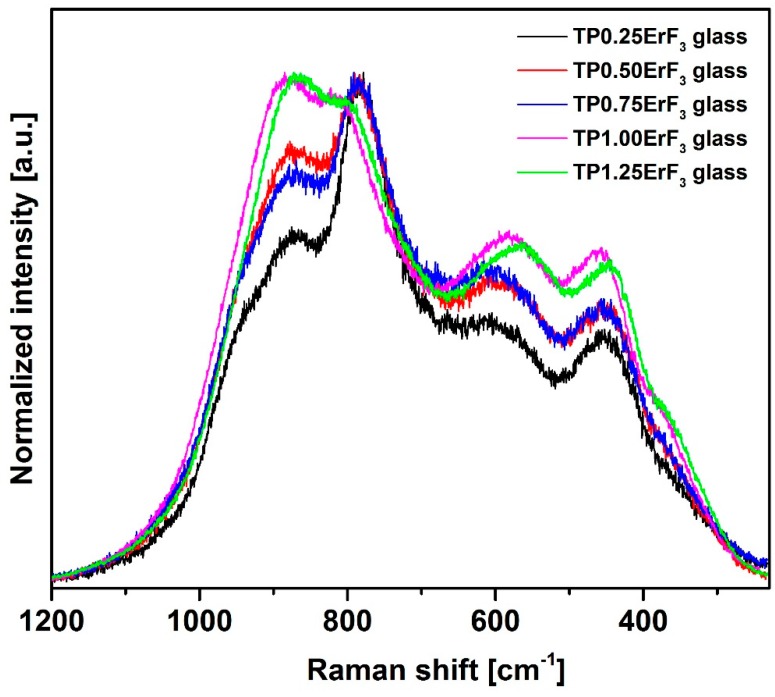
Raman spectra of precursor glass doped with varying ErF_3_ content.

**Figure 9 materials-12-03429-f009:**
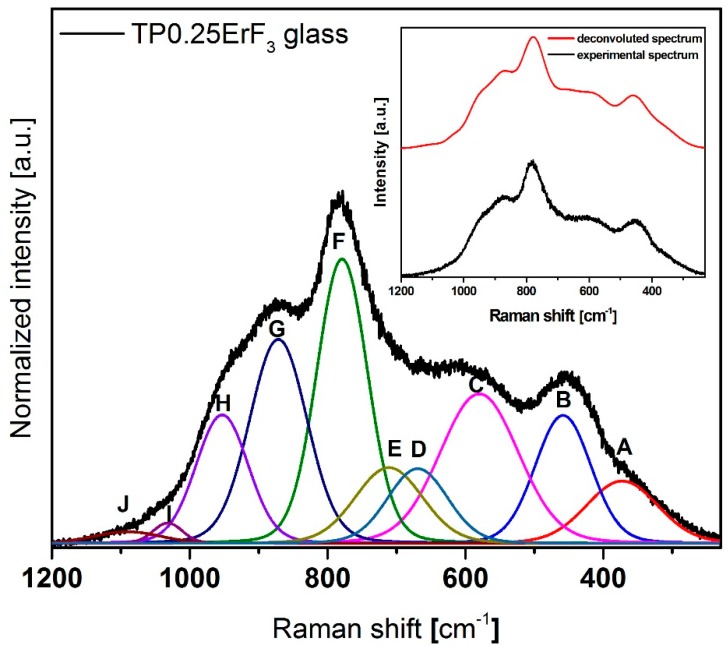
Deconvoluted Raman spectra of TP0.25ErF_3_ glass. (inset) Deconvoluted and experimental spectra.

**Figure 10 materials-12-03429-f010:**
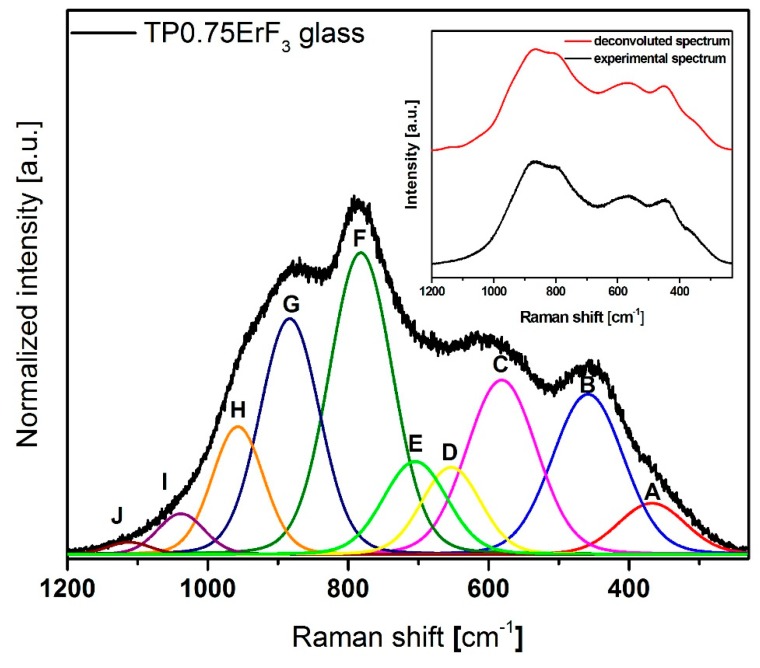
Deconvoluted Raman spectra of TP0.75ErF_3_ glass. (inset) Deconvoluted and experimental spectra.

**Figure 11 materials-12-03429-f011:**
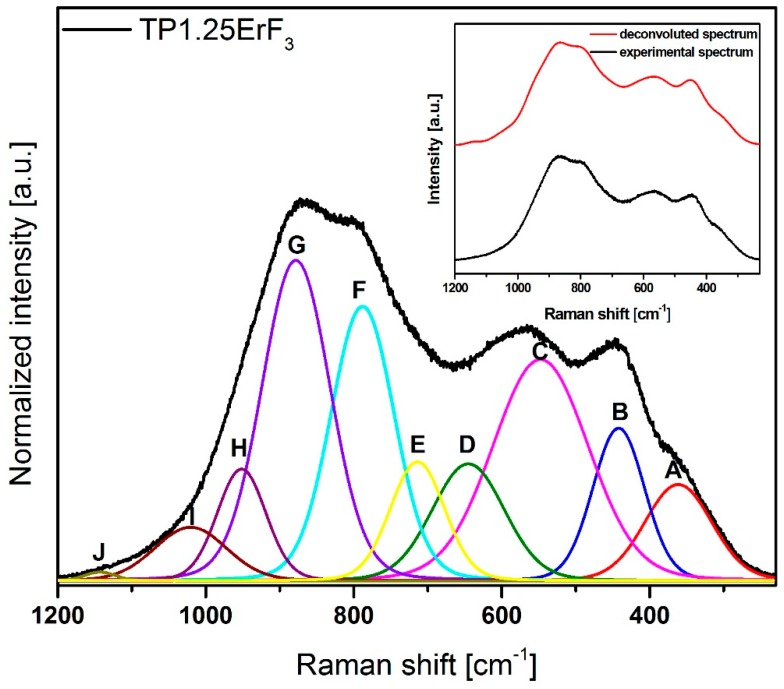
Deconvoluted Raman spectra of TP1.25ErF_3_ glass. (inset) Deconvoluted and experimental spectra.

**Figure 12 materials-12-03429-f012:**
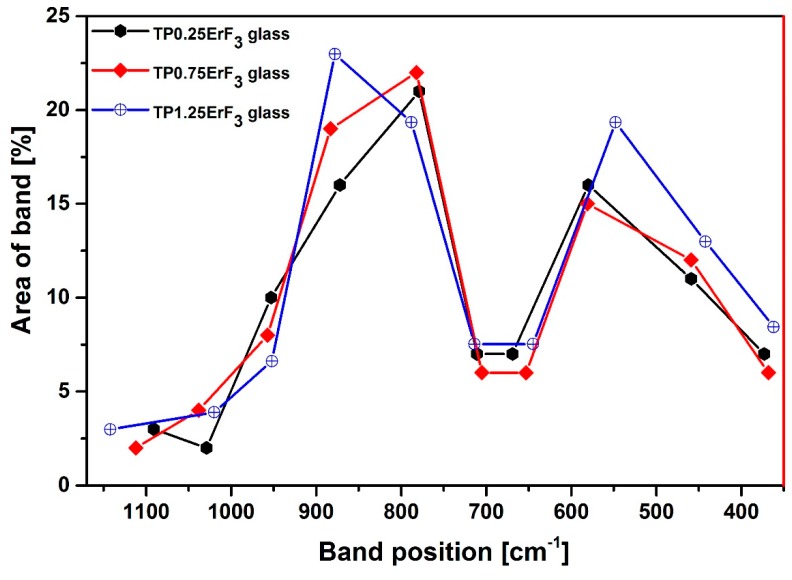
Plot of bands position and their area for deconvoluted Raman spectra of TP0.25ErF_3_, TP0.75ErF_3_, and TP1.25ErF_3_ glasses.

**Figure 13 materials-12-03429-f013:**
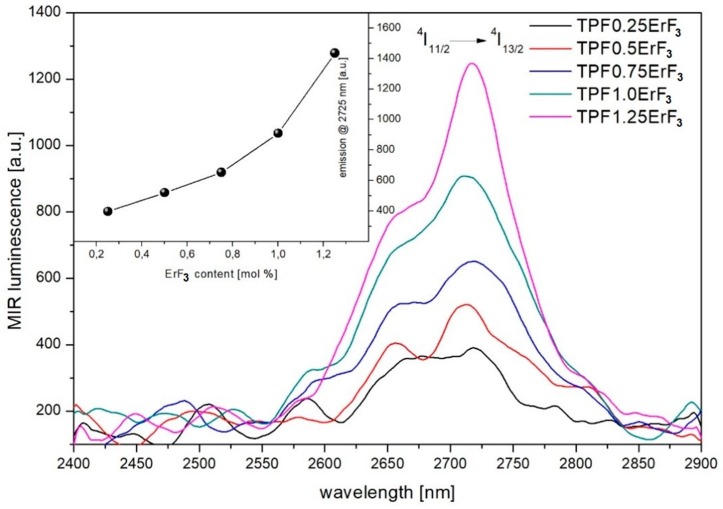
2725 nm emission spectra of oxyfluoride phospho-tellurite glass with varying ErF_3_ content. (inset) The intensity changes of 2725 nm emission band as a function of ErF_3_ concentration.

**Figure 14 materials-12-03429-f014:**
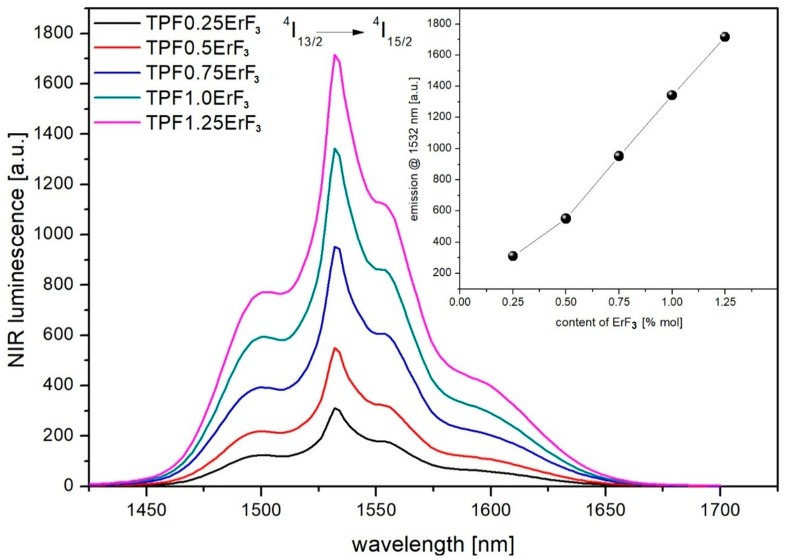
1553 nm emission spectra of oxyfluoride phospho-tellurite glass with varying ErF_3_ content. (inset) The intensity changes of 1553 nm emission band as a function of ErF_3_ concentration.

**Figure 15 materials-12-03429-f015:**
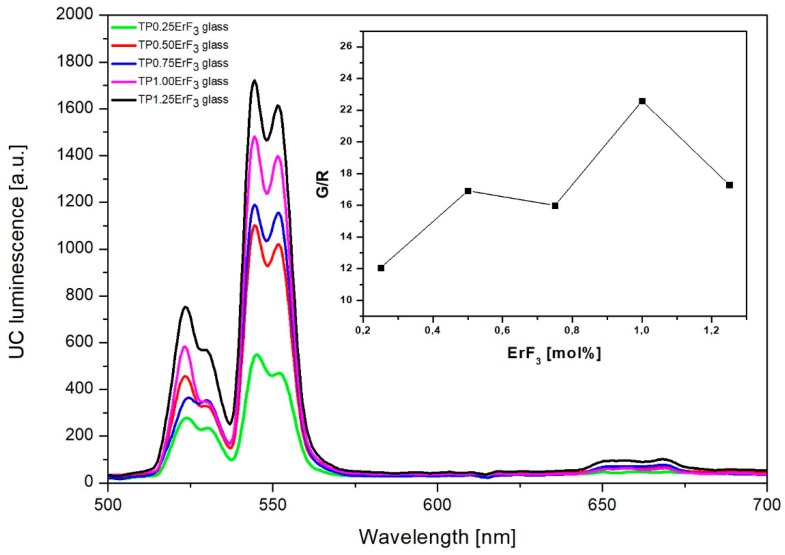
Upconversion emission of phospho-tellurite glass with varying ErF_3_ content. (inset) The intensity changes of green (528 and 551 nm) and red (668 nm) emission bands as a function of ErF_3_ concentration.

**Figure 16 materials-12-03429-f016:**
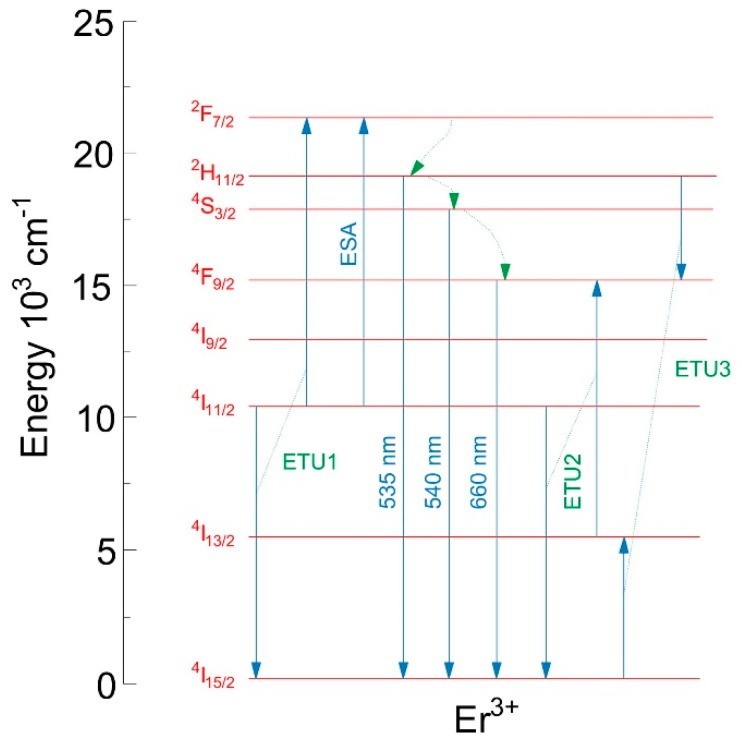
The energy scheme.

**Figure 17 materials-12-03429-f017:**
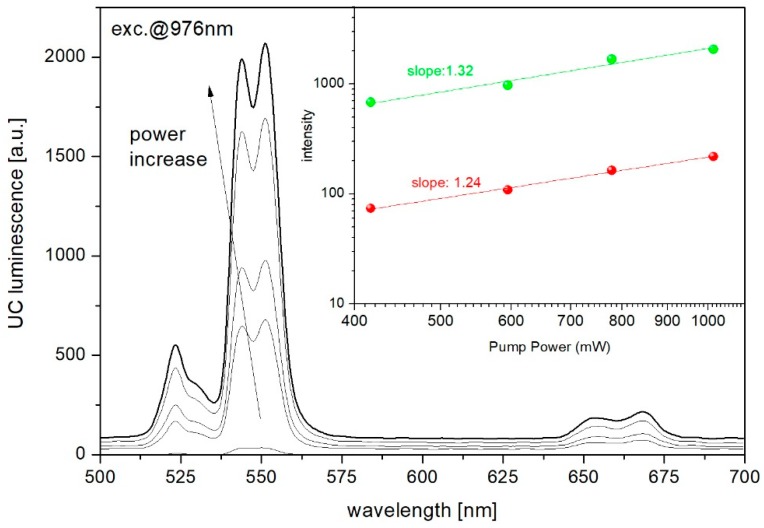
Upconverison luminescence spectra of TP1.25ErF_3_ glass as a function of pumping power. (inset) The log-log dependency of upconversion emission intensity for green and red radiative transitions.

**Figure 18 materials-12-03429-f018:**
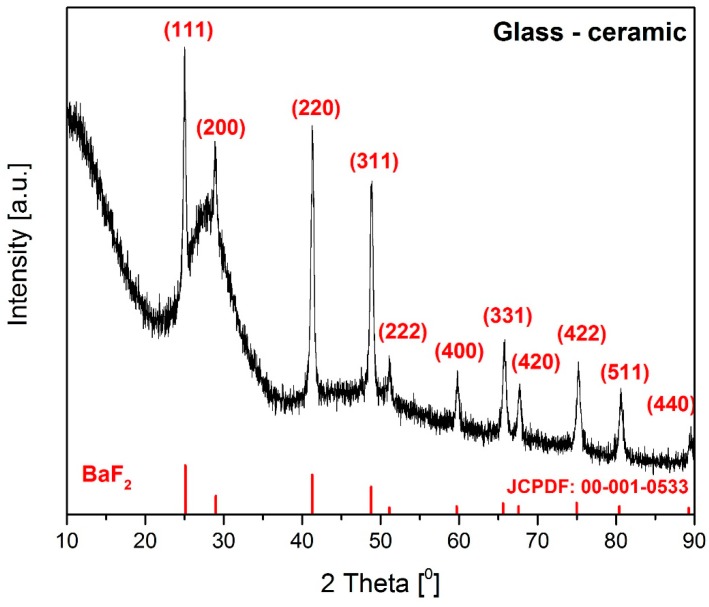
XRD patterns of glass-ceramic.

**Figure 19 materials-12-03429-f019:**
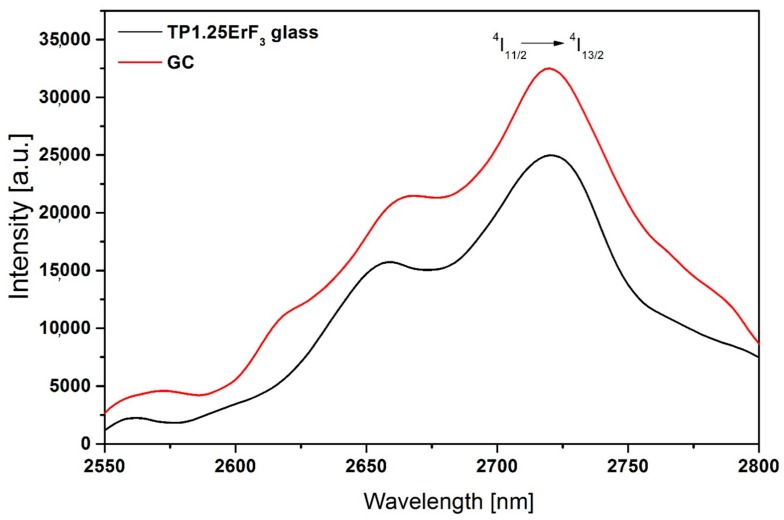
2725 nm emission spectra of TP1.25ErF_3_ glass and glass-ceramic.

**Figure 20 materials-12-03429-f020:**
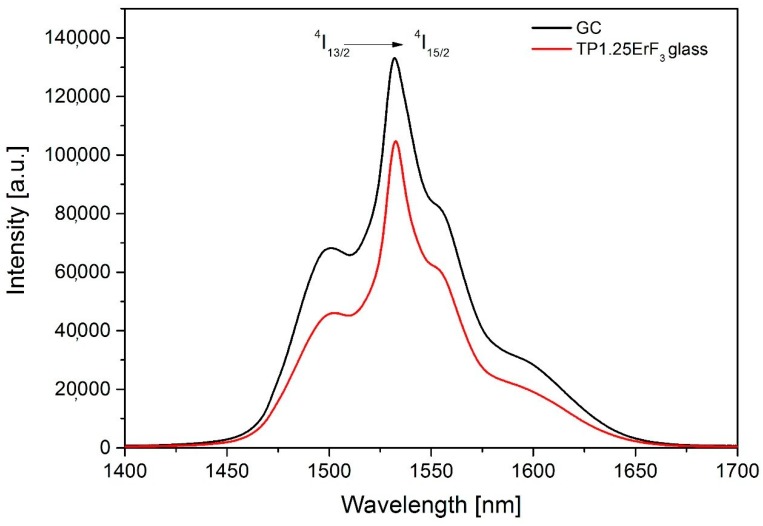
1553 nm emission spectra of TP1.25ErF_3_ glass and glass-ceramic.

**Figure 21 materials-12-03429-f021:**
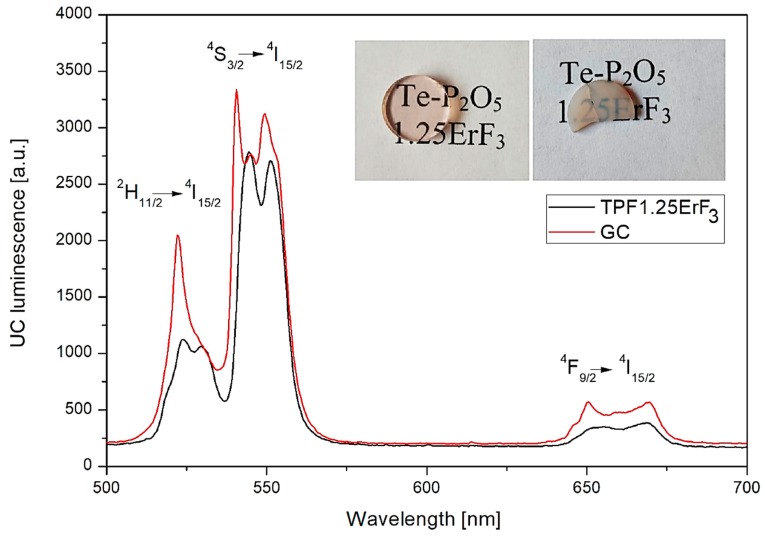
Upconversion emission of TP1.25ErF_3_ glass and glass-ceramic. (inset) Image of fabricated glass and glass-ceramic.

**Table 1 materials-12-03429-t001:** Glass transition (T_g_), crystallization (T_c_), melting (T_m_) temperatures, and thermal stability (ΔT) of erbium-doped phospho-tellurite glass.

Sample of Glass	T_g_ [°C]	T_c_ [°C]	T_m_ [°C]	ΔT [°C]
TP0.25ErF_3_	313 ± 1	438 ± 1	539 ± 1	125 ± 1
TP0.50ErF_3_	320 ± 1	415/458 ± 1	537 ± 1	95 ± 1
TP0.75ErF_3_	322 ± 1	451/483 ± 1	517 ± 1	129 ± 1
TP1.00ErF_3_	324 ± 1	466 ± 1	547 ± 1	142 ± 1
TP1.25ErF_3_	329 ± 1	420/456 ± 1	557 ± 1	109 ± 1

**Table 2 materials-12-03429-t002:** The parameters of bands obtained by deconvolution of the FTIR spectra of TP0.25ErF_3_, TP0.75ErF_3_, and TP1.25ErF_3_ glasses.

**TP0.25ErF_3_ Glass**			
**Band**	**Peak [cm^−1^]**	**Area [%]**	**FWHM [cm^−1^]**
A	545	2 ± 1	15
B	565	3 ± 1	20
C	606	9 ± 2	35
D	674	11 ± 3	50
E	739	8 ± 2	41
F	780	3 ± 1	24
G	802	1 ± 1	16
H	872	3 ± 1	28
I	918	9 ± 2	41
J	979	10 ± 2	30
K	1027	10 ± 2	36
L	1077	17 ± 3	56
M	1135	15 ± 2	63
**TP0.75ErF_3_ Glass**			
**Band**	**Peak [cm^−1^]**	**Area [%]**	**FWHM [cm^−1^]**
A	546	2 ± 1	15
B	567	3 ± 1	20
C	606	9 ± 2	35
D	676	12 ± 3	50
E	738	8 ± 2	41
F	779	3 ± 1	25
G	802	1 ± 1	17
H	873	3 ± 1	29
I	923	9 ± 2	43
J	976	7 ± 2	28
K	1026	14 ± 2	43
L	1094	21 ± 3	58
M	1158	7 ± 2	56
**TP1.25ErF_3_ Glass**			
**Band**	**Peak [cm^−1^]**	**Area [%]**	**FWHM [cm^−1^]**
A	546	3 ± 1	16
B	567	3 ± 1	19
C	606	9 ± 2	36
D	681	11 ± 3	51
E	740	7 ± 2	39
F	779	3 ± 1	24
G	801	1 ± 1	16
H	873	3 ± 1	28
I	924	9 ± 2	43
J	974	4 ± 2	27
K	1024	15 ± 2	48
L	1102	27 ± 4	64
M	1178	4 ± 2	56

**Table 3 materials-12-03429-t003:** Band assignment.

Band [cm^−1^]	Assignment
A 545/546/546	deformation vibration of δO–P–O and δO–P–O bonds in Q^2^ units [68]
B 565/567/567	
C 606/606/607	stretching vibrations of Te–O bonds in TeO_4_ (tbp) units [69,70]
D 674/676/681	
E 739/738/740	stretching vibrations of TeO_3_ (tp) units or TeO_3+1_ polyhedra [70]
F 780/779/779	the vibration of the continuous network composed of TeO_4_ and Te–O stretching vibration of TeO_3+1_ polyhedra [69] or symmetric P–O = P bonds in Q^1^ units [71]
G 802/802/801	asymmetric stretching vibrations of TeO_3_ (tp) units or TeO_3+1_ polyhedral [69,72]
H 872/873/873	asymmetric stretching vibrations of Q^2^ units [73]
I 918/923/924	asymmetric stretching vibrations of P–O–P linked with metaphosphate chainand P–F groups in Q^2^ units [74,75]
J 979/976/974	asymmetric stretching vibrations of P–O^−^ bonds in Q^0^ units [68]
K 1027/1026/1024	symmetric stretching vibrations of PO^2−^_3_ groups in Q^1^ units [74]
L 1077/1094/1102	asymmetric stretching vibrations of PO^2−^_3_ groups in Q^1^ units [76]
M 1135/1158/1178	asymmetric stretching vibrations of non-bridging oxygen in Q^2^ units [77]

**Table 4 materials-12-03429-t004:** The parameters of bands for the deconvoluted Raman spectra of TP0.25ErF_3_, TP0.75ErF_3_, and TP1.25ErF_3_ glasses.

**TP0.25ErF_3_ Glass**			
**Band**	**Peak [cm^−1^]**	**Area [%]**	**FWHM [cm^−1^]**
A	373	7 ± 2	69
B	459	11 ± 2	54
C	580	16 ± 3	76
D	669	7 ± 2	58
E	711	7 ± 2	65
F	779	21 ± 4	49
G	872	16 ± 2	56
H	953	10 ± 3	51
I	1029	2 ± 1	28
J	1091	3 ± 0.5	56
**TP0.75ErF_3_ Glass**			
**Band**	**Peak [cm^−1^]**	**Area [%]**	**FWHM [cm^−1^]**
A	368	6 ± 2	64
B	459	12 ± 2	67
C	581	15 ± 3	66
D	653	6 ± 2	56
E	704	6 ± 2	60
F	783	22 ± 4	59
G	883	19 ± 2	56
H	957	8 ± 3	49
I	1038	4 ± 1	44
J	1112	2 ± 0.5	40
**TP1.25ErF_3_ Glass**			
**Band**	**Peak [cm^−1^]**	**Area [%]**	**FWHM [cm^−1^]**
A	361	7 ± 2	60
B	442	12 ± 2	47
C	548	19 ± 3	86
D	645	5 ± 2	65
E	714	7 ± 2	49
F	788	19 ± 4	57
G	878	23 ± 2	61
H	952	5 ± 3	44
I	1020	2 ± 1	65
J	1114	1 ± 0.5	28

**Table 5 materials-12-03429-t005:** Raman band assignment.

Band [cm^−1^]	Assignment
A 373/368/361	bending vibration of Te–(O, F)–Te or O,F–Te–O,F bands of [Te(O, F)_4_] trigonal bipyramidal units [81,82]
B 459/459/442	
C 580/581/548	
D 669/653/645	stretching variation of Te–O,F bonds in [Te(O,F)_4_] units [83]
E 711/704/714	Bands assigned to the Te(O,F)_4_ tbp units [84]
F 779/783/788	Te–O^−^ stretching vibration in [TeO_3_] trigonal pyramids or symmetric stretching vibration in [TeO_3+1_] units [85]
G 872/883/878	symmetric stretching vibration of the P–F bonds [86]
H 953/957/952	symmetric stretching vibration of PO_4_ in Q^0^ units [87]
I 1029/1038/1020	stretching vibrations of bridging P–O–P bonds in Q^1^ units [25,90]
J 1091/1112/1142	stretching vibrations of non-bridging bonds PO_2_ of Q^2^ units [23,25,90]

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
