# Peer review of "Spectroscopic Properties of Erbium-Doped Oxyfluoride Phospho-Tellurite Glass and Transparent Glass-Ceramic Containing BaF2 Nanocrystals"

_materials, 2019, doi:10.3390/ma12203429_

Round 1

Reviewer 1 Report

The research work entitled "Spectroscopic Properties of Erbium-Doped Oxyfluoride Phosphor-Tellurite Glass and Transparent Glass-Ceramic Containing BaF2 Nanocrystals" by M. Lesniak et al describes the effect of ErF3 addition on thermal, structure and spectroscopic properties of oxyfluoirde phosphor-tellurite glasses. While the manuscript can potentially be interesting, it does require some consideration and analysis, prior to its eventual publication.

1)     The “Tb3+” in line 34 should be “Yb3+”.

2)     The alphabetic flag in Fig. 4 to Fig. 6 are not consistent with the following table.

3)     Could the author give some detailed description of the photoluminescence test? For example, the pumping and emission collecting direction.

4)     Why the upconversion emission can be obtained under the excitation of 455 nm (descripted in line 302)? And, could the author explain why “the ratio of green to red increased gradually with the increase of ErF3” (descripted in line 308-310)? As I know, the ratio usually decreased due to the cross-relaxation between Er3+ ions in high doping concentrations.

5)     It would be better to show the curve of JCPDF: 00-001-0533 in Fig. 12 for comparison. And, the transmittance spectra or the photograph of the specimen should be present here to confirm the glass-ceramic is transparent.

6)     How the value “74%” in line 378 was obtained? As shown in the Fig. 13, the emission intensity of Er3+ ions from glass-ceramic is about 3 times compared with the emission from as-prepared glass.

7)     For MIR and NIR emissions (Fig.7 and Fig. 8), why the intensity slightly increases when doping concentration lower than 0.75 mol% and significantly increases when the doping concentration increased to 1.0 mol%?

8)     As we know, the emission intensity can be easily influenced by the external conditions (specimen size, pumping method, position, direction et al.). Did the author measure the decay curves of Er3+ ions in glass-ceramic and as-prepared specimens? It would be much better to explain the incorporation of Er3+ ions into the BaF2 nanocrystals. (or TEM images combined with element mapping result. Maybe this is difficult for the tellurite based glasses.)

9)     There are some errors occurred in line 469 and line 485 for the description of doping concentration of ErF3. Similar mistakes occurred several times in this manuscript. It would be better to have a carful check and correction.

10)   There are some typos and grammatical errors that should be fixed.

Reviewer 2 Report

In conclusion. :

            The article can be published in the present form with some corrections and explanations as described bellow:

It is not very clear how a such a small amount (i.e. 1.25 mol%) can produce a suc a big changes in thermal behavior. Also, how Q3 tetrahedra can be present in this series of glasses, for such a low amount( 10 mol%) of P2O5 which exists in the glass. H band, infrared absorption can present and both vibration are possible.

            4. Why in the emission curve at 1550 nm the intensity is higher only with 20% than for the glass, but in the situation of up conversion, the intensity of the line is 20 k,  much higher in the glass ceramic

Reviewer 3 Report

The paper deals with the preparation and characterization of glasses and glass-ceramics in the system (40-x)TeO2-10P2O5-15 45(BaF2-ZnF2)-5Na2O-xErF3. Remarkable results are shown, regarding the enhanced luminescent properties of the glass-ceramic, when comparing with the starting glass. This piece of work has grate potential to be published in Materials. However the following points should be addressed before publication:

English language must be completely revised. There is a series of misleading or imprecise sentences that makes the text hard to follow. Some examples are listed below:

line 33-36 - Long sentence.

line 40 - "...co-dopants, and besides the glassy matrix provided high..."

line 67 - "...control the crystal size during obtain glass-ceramic."

line 69-70 - "...oxyfluoride phosphate glass-ceramic is special interest as hosts for luminescent rare-earth ions"

line 104 - "After obtaining the chemical and thermal stability glass,..."

line 162-163 - "For oxide tellurite glasses incorporating additional network modifier such as alkali and alkaline-earth metals, showed an increase in the glass transition..."

line 171 - "Effect of erbium-doped tellurite glasses on thermal properties was studied in the literature."

line 179 - "...may led to increase Tg value..."

line 222 - "...FTIR spectra were carried out on..."

line 225 - "...ErF3 concentration in the 1300-500 cm-1..."

line 229 - "Based on Fig. 3 might be concluded that..."

line 341-343 – confusing sentence.

Line 345 – “However, on cadmium fluoride CdF2 and lead fluoride PbF2 are poisonous raw materials,...”

Line 349-350 – “Next researchers studied systemically preparation of...”

Line 361 – “ions like Er3+ are considered as luminescence centers…”

Line 370-371 -  “The erbium-doped germano‐gallate oxyfluoride glass‐ceramics containing BaF2 nanocrystals high transmittance in the mid‐infrared region were prepared...”

Line 376 – “...the 2725 nm emission intensity of glass-ceramic is higher than TP1.25ErF3 glass.”

Line 378 – “...for 3h time of annealing the 74% enhancement was…”

Line 382-383 – Reformulate.

Line 391-393 – “The MIR, NIR and up-conversion emission spectra of Er3+ in the obtained glass-ceramic were much stronger than that in the TP1.25ErF3 because the Er3+ incorporated the crystalline environment of BaF2 nanocrystals”

Line 398 – “...have low phonon energies than oxide glasses…”

Line 419 – “Based on thermal stability values was found that oxyfluoride phospho-tellurite glass...”

Line 425 – “I was found, that…”

Line 436-437 – “Presence of tellurium and phosphorus cations in the glass network attract fluoride ions yield a competition between them.”

Line 443-446 – Long and confusing sentence.

Line 449 – “...from the log-log dependence of emission intensity on the excitation power indicated,...”

Line 453 – “...of the emission peaks is found to increase with increasing the mol% ErF3 in the precursor glass.”

Line 456 – “By annealing for 3 hours in 400 ºC of the precursor oxyfluoride glass doping with the highest value...”

Line 470 – “The data presented in this paper firstly report of transparent Er3+‐doped BaF2 glass-ceramic in...”

Line 471 – “...In the following step detail characterization of...”

Line 478 – “Increased in glass transition 478 temperature with the increase of ErF3 concentration...”

Line 483 – “...was found to increase with increasing the mol% of ErF3 in the precursor glass.”

Many occurrences: "...in the room temperature..."; wrong use of singular/plural.

In the introductory section the authors state that “to obtain the optimum dopant concentration in the nanocrystals and thus the highest luminescence efficiency, precise control of the rare-earth concentration and crystallization process is required.”. However, in the experimental section it is not clear if optimizations were performed in order to obtain the desired glass-ceramic. line 71. The authors should mention the network-former mixing effect when speaking about the improvement of tellurite glass properties by the adition of P2O5 glass former. One issue in oxyfluoride glasses is the omnipresent fluoride losses during the melting step, which can lead to considerable spectroscopic consequences. Even in systems were the mass losses during melting were not significant, grate fluorine losses were detected by 19F quantitative solid state NMR (Gonçalves et al. Materials Chemistry and Physics 157 (2015) 45-55). Did the authors quantify the mass losses during melting? If yes, the values should be mentioned in the experimental section. Line 168: definition of the stability factor (ΔT) should be given at this point. Line 202 – The most widely used abbreviation for “Full Width at Half Maximum” is FWHM and not FWTH as used in the manuscript. Lines 205-216. Here the authors compare values of Tg, Tm and ΔT for the various glasses. The precise determination of Tg values by the onset method is sometimes rather difficult. Also, in the present study, the presence of multiple crystallization peaks makes it hard do determine precisely the Tc values, specially for sample TP1.00ErF3, where there seems to be two partially overlapping contributions (assigned as Tc1 and Tc2 for the samples with close composition). For these reasons the authors must insert error values in Table 1 and related text. Only with these error margins the authors are able to make fair comparison with the literature. Also, in this same paragraph, there are some wrong symbols (division symbol instead of hyphen and number symbol instead of degrees symbol). line 228. “The FTIR spectra were normalized to the band 225 at ~ 1050 cm−1.” Why was this band chosen for normalization of the spectra? Line 234. Figures 4-6 show the deconvolutions, and not 3-5 as stated in the manuscript. Figures 4-6: I have concerns about such detailed deconvolutions of these broad FTIR spectra. From this data a variety of deconvolution strategies would lead to the same result. The author should mention which kind of constraints they have used for the decolvolutions (if any). Also regarding FTIR deconvolutions, from the data presented it is not possible to compare deconvoluted and experimental data, the authors should use different colors for the total deconvoluted spectrum and the experimental one, or alternatively, stack experimental and deconvoluted spectra. Peak positions in the text and in Table 2 do not correspond to the labels in the FTIR spectra in Figures 4-6, they seem to be inverted. Also, errors should be presented for the FTIR deconvolution parameters and ambiguities in the deconvolution must be addressed. The areas presented in Table 2 are presented in arbitrary units, which is counter intuitive. These areas would be better represented as fractions (% of total area). Also, comparing absolute intensities among samples is not the best approach, since the used KBr pallet method is not completely quantitative. Representation of the areas as percentage-area will make the comparison more consistent. A plot of peak positions and intensities, including error bars, would help the reader to follow the variations. Line 260-263. This paragraph is confusing and probably over-interpret the FTIR data. The formation of F-P-F linkages is not well explained and the citation of Ref. 71 is not well justified, since in this reference pure oxide glasses were studied. Figures 7-9: please refer to the inset in the figure caption. Figure 10 caption must be reformulated, it doesn’t describe the Figure completely. Also, only the significant digits of the fitting parameters must be shown in the inset. Finally, the spectrum with lowest intensity is partially hidden by the axis. Lines 312-314: the conclusion in this sentence should be better explained for the non-expert reader. Maybe a short citation about what is the range expected for the slopes for a two-phonon mechanism. Line 333: Please add reference to the Scherrer’s formula. Line 334: Add error for the BaF2 crystalline lattice parameter. After taking into consideration the comments regarding the FTIR analyses, the authors should revisit the conclusions about phosphate speciation. If possible, the authors should consider the performance of 19F solid-state NMR to confirm or discard the presence of P-F bonds and their variation with composition. 

Round 2

Reviewer 1 Report

The authors has correct many mistakes follow my suggestions. But, I really cannot believe the correct of some data directly.

For example, in my previous comments,  I'm not sure how the 74% was obtained. The author just changed the description to "three times".

And, same response also appeared in response to the upconversion of "green to red ratio" and "increase of NIR intensity with doping concentration of Er3+ ions".

Although the author has pay lots of effort to response the comment, please response or explain the mistake again,and do not just correct as what I suggested:

Could the author explain “the ratio of green to red increased gradually with the increase of ErF3” (descripted in line 308-310)? Even you changed the inset curve, but the ratio of G/R from 0.25 Er3+% is absolutely not right in macroscopic.   How the value “74%” in line 378 was obtained? Just mistake?   For MIR and NIR emissions (Fig.7 and Fig. 8), why the intensity slightly increases when doping concentration lower than 0.75 mol% and significantly increases when the doping concentration increased to 1.0 mol%?

Thank you.

Reviewer 3 Report

The authors have corrected many of my and the other referees corrections and improved the manuscript considerably. However, I still have some concerns, as follows:

The English language still need further revision.

The thermal stability factor DT is indeed explained in the text. However, the definition should be given at the first appearance in the text.

The authors followed my suggestion and provided error values for the characteristic temperatures from DSC curves. However, the given errors are not realistic. The determination of the Tc temperature from the sharp peak for the 25ErF3 glass cannot have similar error to the determination from the wide plateau observed for the 1.00ErF3 glass.

Still, in the DSC section (lines 230-241), many incorrect symbols were left uncorrected.

Regarding the FTIR deconvolutions, the coefficient of determination (R) is not the only parameter to be considered for the error estimation in the deconvolution of such broad spectra. Accuracy in the final fitting doesn’t mean that the model is correct. For this reason, it is advisable to make error estimative based on small variations of the deconvolution parameters. For example, small variation in position and linewidth for one component can be compensated by variations in neighboring components, giving fittings with similar R values. For this reason, error estimative for line positions and areas should be given.

Serious conclusions are taken from this unconstrained FTIR fitting, such as the formation of F-P linkages. It is said that the shift of the bands at ~ 1058 cm-1 and ~ 1114 cm-1 with increasing ErF3 indicates the formation of F-P linkages. However, comparison of the left side of the FTIR spectra in Figure 3 show that they are very similar, and the variation in position doesn’t correlate so well with the composition. An evaluation of the errors involved in these fittings, or effect of constraints, such a linewidth or line positions, would help to confirm such conclusions.

Also, why the authors didn’t fit all the available FTIR spectra? This would give more experimental evidence to support the conclusions.

Round 3

Reviewer 1 Report

Agree